# Tuning microtubule dynamics to enhance cancer therapy by modulating FER-mediated CRMP2 phosphorylation

Yiyan Zheng[1,2], Ritika Sethi[3], Lingegowda S. Mangala[4,5], Charlotte Taylor[1,2], Juliet Goldsmith[1,2], Ming Wang[1,2], Kenta Masuda[1,2], Eli M. Carrami [1,2], David Mannion[1,2], Fabrizio Miranda [1,2], Sandra Herrero-Gonzalez[1,2], Karin Hellner[1,2], Fiona Chen[1,2], Abdulkhaliq Alsaadi[1,2], Ashwag Albukhari[1,2,6], Donatien Chedom Fotso[1,2], Christopher Yau[7,8], Dahai Jiang[4,5], Sunila Pradeep[4], Cristian Rodriguez-Aguayo[5,9], Gabriel Lopez-Berestein[5,9], Stefan Knapp[3,10], Nathanael S. Gray [11,12], Leticia Campo[13], Kevin A. Myers[13], Sunanda Dhar[14], David Ferguson [15], Robert C. Bast Jr.[9], Anil K. Sood[4,5], Frank von Delft [3,16,17] & Ahmed Ashour Ahmed[1,2]

Though used widely in cancer therapy, paclitaxel only elicits a response in a fraction of patients. A strong determinant of paclitaxel tumor response is the state of microtubule dynamic instability. However, whether the manipulation of this physiological process can be controlled to enhance paclitaxel response has not been tested. Here, we show a previously unrecognized role of the microtubule-associated protein CRMP2 in inducing microtubule bundling through its carboxy terminus. This activity is significantly decreased when the FER tyrosine kinase phosphorylates CRMP2 at Y479 and Y499. The crystal structures of wild-type CRMP2 and CRMP2-Y479E reveal how mimicking phosphorylation prevents tetra-merization of CRMP2. Depletion of FER or reducing its catalytic activity using sub-therapeutic doses of inhibitors increases paclitaxel-induced microtubule stability and cytotoxicity in ovarian cancer cells and in vivo. This work provides a rationale for inhibiting FER-mediated CRMP2 phosphorylation to enhance paclitaxel on-target activity for cancer therapy.

[1] Ovarian Cancer Cell Laboratory, Weatherall Institute of Molecular Medicine, University of Oxford, Headington, Oxford OX3 9DS, UK. [2] Nuffield Department of Obstetrics & Gynaecology, University of Oxford, Women's Centre, John Radcliffe Hospital, Oxford OX3 9DU, UK. [3] Structural Genomics Consortium, Nuffield Department of Medicine, University of Oxford, Old Road Campus Research Building, Oxford OX3 7DQ, UK. [4] Department of Gynecologic Oncology, The University of Texas MD Anderson Cancer Center, 1515 Holcombe Blvd, Houston, TX 77030, USA. [5] Center for RNAi and Non-Coding RNA, The University of Texas MD Anderson Cancer Center, 1515 Holcombe Blvd, Houston, TX 77030, USA. [6] Biochemistry Department, Faculty of Science, King Abdulaziz University, Jeddah 21551, Saudi Arabia. [7] Wellcome Trust Centre for Human Genetics and NIHR Biomedical Research Centre, Roosevelt Drive, Oxford OX3 7BN, UK. [8] Department of Statistics, 1 South Parks Road, Oxford OX1 3TG, UK. [9] Department of Experimental Therapeutics, University of Texas MD Anderson Cancer Center, Houston, TX, USA. [10] Goethe-University Frankfurt, Institute for Pharmaceutical Chemistry and Buchmann Institute for Life Sciences, Riedberg Campus, Frankfurt am Main 60438, Germany. [11] Department of Biological Chemistry and Molecular Pharmacology, Harvard Medical School, Boston, MA 02115, USA. [12] Department of Cancer Biology, Dana-Farber Cancer Institute, Boston, MA 02215, USA. [13] Department of Oncology, University of Oxford, Old Road Campus Research Building, Roosevelt Drive, Oxford OX3 7DQ, UK. [14] Department of Histopathology, Oxford University Hospitals, Oxford OX3 9DU, UK. [15] Nuffield Division of Clinical Laboratory Sciences, Radcliffe Department of Medicine University of Oxford, Oxford University Hospitals, Oxford OX3 9DU, UK. [16] Diamond Light Source Ltd, Harwell Science and Innovation Campus, Didcot OX11 0QX, UK. [17] Department of Biochemistry, University of Johannesburg, Auckland Park 2006, South Africa. Correspondence and requests for materials should be addressed to A.A.A. (email: ahmed.ahmed@obs-gyn.ox.ac.uk)

Paclitaxel increases microtubule polymerization and stability, induces mitotic arrest, promotes cancer cell death, and improves patient survival[1,2]. However, paclitaxel only elicits a response in a fraction of patients[3]. Poor pharmacokinetics represents a major obstacle for appropriate drug delivery. Therefore, it is likely that paclitaxel interacts with microtubules in cancer cells at sub-stoichiometric concentrations[4]. Alternative formulations have already shown promise in enhancing paclitaxel pharmacokinetics and, thereby, its efficacy, highlighting the need for overcoming sub-optimal on-target activity[5].

Previous work has shown that microtubule dynamics of cancer cells have a profound effect on the magnitude of paclitaxel response. For example, mutations in the non-binding sites for paclitaxel in α-tubulin or β-tubulin result in an increase in microtubule dynamic instability and significant paclitaxel resistance[6,7]. Similarly, overexpression of βIII-tubulin which is associated with an increase in dynamic instability has been shown to correlate with clinical resistance to paclitaxel[8]. We and others have previously shown that the state of microtubule stability in cancer cells prior to treatment is an important determinant of the magnitude of paclitaxel-induced microtubule stabilization[9–12]. However, whether microtubule dynamics manipulation can be exploited for enhancing paclitaxel cytotoxicity has remained untested.

Collapsin response mediator protein 2 (CRMP2) is an intracellular phosphoprotein that plays an important role in regulating cytoskeletal dynamics. It forms a tetramer that interacts with tubulin[13] or pre-formed microtubules[14] to modulate important microtubule functions such as neuronal axonal growth and axon-dendrite fate determination[15]. It is thought to influence microtubule assembly via its interactions with αβ-tubulin heterodimers[13]. Several key kinases are reported to phosphorylate CRMP2 at the carboxy terminus, thereby inhibiting its binding activity to tubulin. For example, serine/threonine phosphorylation of CRMP2 via glycogen synthase kinase 3β (GSK3β) or cyclin-dependent kinase 5 (CDK5) interferes with the ability of CRMP2 to bind to tubulin[16–19]. However, whether or not CRMP2 can modulate microtubule bundling and stability and whether such functions are regulated by phosphorylation has not been previously established.

Here, we report that FER phosphorylates CRMP2 at Y479 and Y499, resulting in a significant structural change of CRMP2 that reduces its ability to induce microtubule bundling. Inactivation of FER increases microtubule stability in ovarian cancer cells, potentiating the cytotoxicity of paclitaxel. Our data identify FER tyrosine kinase as a plausible target for therapy that acts in synergy with one of the most commonly used chemotherapeutic drugs in the treatment of ovarian cancer.

## Results

**FER phosphorylates CRMP2 at Y479 and Y499.** We set out to identify a microtubule-associated protein whose function: (1) is modulated by an oncogene kinase that could be targeted therapeutically and (2) directly controls microtubule bundling and stability. FER kinase has been reported to promote ovarian cancer metastasis[20]. Its expression level has been previously correlated with the level of microtubule stability[12]. In addition, previous reports found that FES, the only other family member of FER, is able to phosphorylate CRMP2 at Y32[21]. CRMP2 has been reported to be a known modulator of microtubule assembly[13]. We, therefore, tested whether CRMP2 can modulate microtubule bundling and stability and whether this function can be regulated by FER. In vitro kinase assays using a catalytically active glutathione S-transferase (GST)-tagged truncation of FER (residues 541–822) and recombinant CRMP2 (residues 13–516) confirmed that FER phosphorylates CRMP2 (Supplementary Fig. 1a).

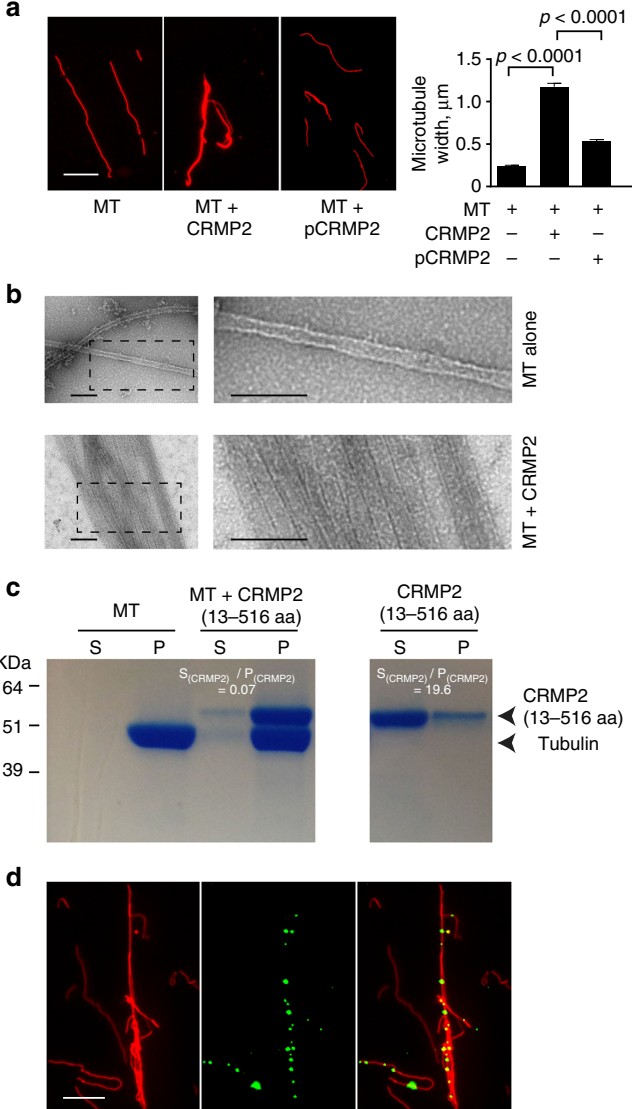

**Fig. 1** FER phosphorylation of CRMP2 impairs its microtubule bundling activity. **a** Paclitaxel-stabilized rhodamine-labeled microtubules were incubated in the absence or presence of CRMP2 or pCRMP2 (phosphorylated by FER kinase) at room temperature for 40 min before fluorescence microscopy was performed and images were obtained. Bar plots represent the mean + s.e.m. of microtubule width in μm from at least 150 individual microtubules per condition tested. Data presented are typical of at least three independent replicates. Scale bar is 10 μm. **b** Electron microscopy images of microtubules in the absence or presence of recombinant 6His-CRMP2. The dashed rectangles in the left panel are the areas zoomed in on the right panel. Scale bar is 100 nm. **c** Recombinant 6His-CRMP2 (residues 13–516) was purified and incubated with paclitaxel-stabilized microtubules for 30 min at room temperature. Subsequently, the samples were ultracentrifuged at 200,000×g for 30 min before analysis using SDS-PAGE and coomassie staining. S: supernatant, P: pellet. **d** Paclitaxel-stabilized rhodamine-labeled microtubules were incubated in the absence or presence of CRMP2 at room temperature for 40 min. CRMP2 localization was revealed by anti-CRMP2 antibody. Scale bar is 10 μm

Importantly, using purified recombinant CRMP2 and tubulin that was purified from pig brains (Supplementary Fig. 1b), we found that FER phosphorylation of CRMP2 significantly reduced its microtubule polymerizing activity as evidenced by light scatter assays (Supplementary Fig. 1c). Using total internal reflection

fluorescence microscopy (TIRFM) to monitor the elongation of individual microtubules, we found that pCRMP2 (phospho-CRMP2) had a significant impairment in its ability to induce microtubule elongation (Supplementary Fig. 1d and Supplementary movie 1–3). In addition, pre-formed, paclitaxel-treated, rhodamine-labeled single microtubules were incubated with recombinant CRMP2 proteins and observed using fluorescence microscopy. This revealed that CRMP2 induced a significant increase in the width of microtubules (Fig. 1a), an observation that was akin to microtubule bundling induced by paclitaxel[1]. Electron microscopy confirmed that CRMP2 induced significant re-organization of microtubules in sheet structures composed of individual parallel microtubules with frequent overlap between the sheets (Fig. 1b and Supplementary Fig. 1e). In addition,

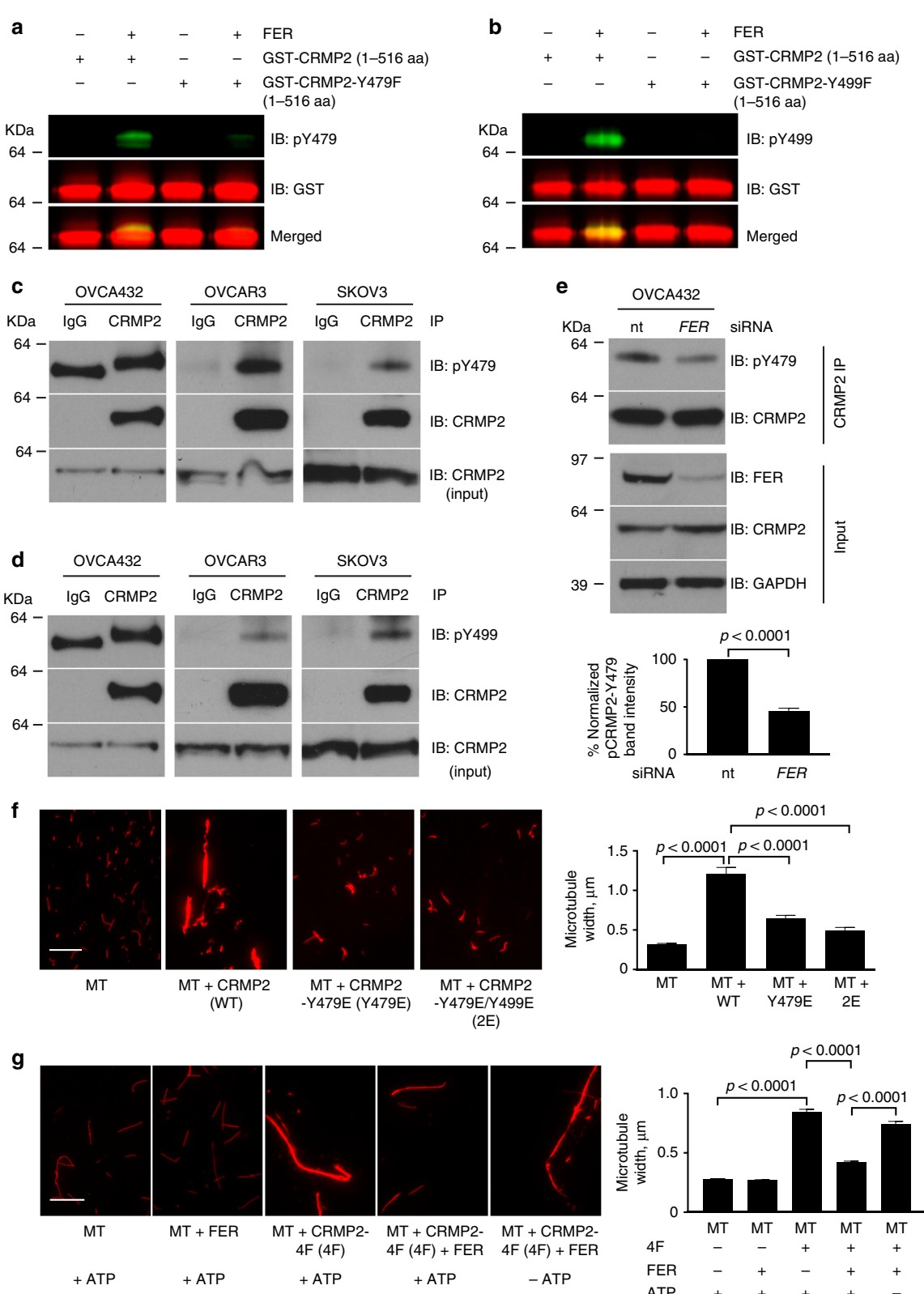

in vitro co-sedimentation assays showed that most of CRMP2 (residues 13–516) co-sedimented with microtubules in the pellet fraction (P). In contrast, most of CRMP2 remained in the supernatant (S) in the absence of microtubules (Fig. 1c) supporting the notion that CRMP2 was able to directly bind to microtubules. Furthermore, CRMP2 co-localized with paclitaxel-stabilized, rhodamine-labeled microtubules as shown by fluorescence microscopy (Fig. 1d). Thus, we provide strong evidence that CRMP2 directly induces microtubule bundling. Since we used paclitaxel-stabilized microtubules, our observation suggests that CRMP2-induced bundling is independent of tubulin assembly. Importantly, FER-induced phosphorylation of CRMP2 significantly reduced its ability to bundle microtubules (Fig. 1a). Taken together, our data show that FER has a profound negative impact on the ability of CRMP2 to induce microtubule polymerization, elongation, and bundling.

To identify FER phosphorylation sites on CRMP2, we performed phosphopeptide mapping using liquid chromatography mass spectrometry (LC-MS) on recombinant CRMP2 with or without prior incubation with FER and identified six phosphorylation sites: Y32, Y251, Y275, Y431, Y479, and Y499 (Supplementary Fig. 2a). Using recombinant full-length CRMP2 protein (residues 1–572) and its truncated versions, residues 13–490, 13–516, 13–526, 13–536, 13–546, and 13–556, we found that the carboxy terminus of CRMP2 is important for its microtubule bundling activity (Supplementary Fig. 2b, c). We therefore focused our further analysis on Y479 and Y499 phosphorylation sites. Phosphotyrosine-specific antibodies for both sites were raised and these confirmed that FER phosphorylated CRMP2 at Y479 and Y499 in vitro (Fig. 2a, b). Importantly, FER was unable to phosphorylate the Y479F or the Y499F mutants of CRMP2, indicating the specificity of the antibodies in vitro (Fig. 2a, b). The validity of the phosphorylation events was further confirmed in vitro by showing that the phosphorylation signal was depleted in a dose-dependent manner following treatment using a FER/FES inhibitor[22] (Supplementary Fig. 3a). Further analysis revealed that FER phosphorylated CRMP2 in ovarian cancer cells since FER inhibition or depletion decreased total CRMP2 tyrosine phosphorylation in the ovarian cancer cell lines OVCA432 or SKOV3 (Supplementary Fig. 3b, c). In addition, using immunoprecipitation of endogenous CRMP2 followed by immunoblotting using phospho-antibodies, we confirmed that Y479 and Y499 were both phosphorylated in multiple ovarian cancer cell lines (Fig. 2c, d). Importantly, the depletion of FER (Fig. 2e) or its inhibition (Supplementary Fig. 3d) in ovarian cancer cells reduced the level of pCRMP2-Y479. Moreover, FER expression level strongly correlated ($r^2 = 0.924$) with pY479-CRMP2 level in multiple ovarian cancer cell lines (Supplementary Fig. 3e, f). Taken together, our data confirmed that FER phosphorylated CRMP2 at Y479 in ovarian cancer cells.

A previous report showed that YES kinase phosphorylated CRMP2 at Y479[23]. YES was expressed in several ovarian cancer cell lines but hardly detected in OVCA432 (Supplementary Fig. 3g). To test the effect of loss of YES on pY479-CRMP2, we

chose OVCAR3 cell line that had high expression of YES kinase and also expressed FER. The level of pY479-CRMP2 reduced in OVCAR3 cells with the depletion of either YES or FER kinases, compared with control non-targeting short interfering RNA (siRNA) treatment (Supplementary Fig. 3h, i). However, the depletion of both kinases together did not further reduce pY479-CRMP2 (Supplementary Fig. 3h, i). This suggests that both kinases might be in the same pathway.

To understand the potential functional relevance of the identified phosphorylation sites, we generated phosphomimetic mutants of CRMP2 by mutating the tyrosine residues to glutamic acid (E): CRMP2-Y479E and CRMP2-Y479E/Y499E. These mutants significantly reduced the microtubule bundling activity of CRMP2 as compared to CRMP2-WT (Fig. 2f). To confirm that direct phosphorylation of only Y479 and Y499 can impair the microtubule bundling activity of CRMP2, we generated a CRMP2 mutant in which all the other FER tyrosine phosphorylation sites were converted to the non-phosphorylatable residue phenylalanine (F): CRMP2-Y32F/Y251F/Y275F/Y431F and termed it CRMP2-4F. This mutant retained the full bundling activity of the WT CRMP2 (Fig. 2g). We also confirmed that the mutation from Y to F does not affect the secondary structures of the various non-phosphorylatable mutants by performing circular dichroism (CD) spectrophotometry on CRMP2–6F (all six FER phosphorylation sites mutated) and CRMP2 wild type (Supplementary Fig. 3j). However, phosphorylation of CRMP2-4F by FER significantly reduced the bundling activity of CRMP2 (Fig. 2g). Thus, the phosphomimetic mutants of CRMP2 or the direct phosphorylation of Y479 and Y499 significantly impair the microtubule bundling activity of CRMP2.

Since FES kinase, the only other family member of FER kinase, is known to phosphorylate CRMP2 Y32[21], we tested the significance of this phosphorylation on bundling. We generated Y32 phosphomimetic mutant proteins, including CRMP2-Y32E and CRMP2-Y32E-Y479E-Y499E. CRMP2-Y32E proteins had no reduced activity on bundling microtubules, compared with CRMP2 wild-type protein (Supplementary Fig. 4a). This was further confirmed by showing that CRMP2-Y32E-Y479E-Y499E and CRMP2-Y479E-Y499E proteins shared similar activity on bundling microtubules (Supplementary Fig. 4b).

**Structural evidence for disruption of inter-molecular interactions upon phosphorylation of CRMP2.** To understand why phosphorylation of CRMP2 affects its microtubule binding and bundling ability, we solved its crystal structure, and that of its phosphomimetic mutant of the FER phosphorylation sites (Y479 and Y499). Our wild-type CRMP2 structure (PDB ID 5MKV) is also a homo-tetramer as previously reported (residues 13–490; PDB ID 2GSE and 5LXX)[24,25]. Specifically, it is a dimer of dimers[25] (Fig. 3a), and the large buried surface (12,690 A$^2$) suggests that the tetrameric state is biologically relevant. Our structure suggests that the phosphorylation of Y479 would disrupt the CRMP2 tetramer and that of Y499 would impair microtubule binding. Even though neither Y479 nor Y499 are located near the dimerization interface (Fig. 3a, left panel), a close investigation of

**Fig. 2** FER phosphorylates CRMP2 at Y479 and Y499. **a**, **b** Recombinant GST-CRMP2 (residues 1–516), GST-CRMP2-Y479F (residues 1–516), or GST-CRMP2-Y499F (residues 1–516) were purified and used for in vitro kinase assays in the absence or presence of active GST-tagged truncated FER protein (residues 521–822). The samples were analyzed by SDS-PAGE, and immunoblotted using the indicated antibodies. IB immunoblot. **c**, **d** Lysates of the indicated ovarian cancer cell lines were immunoprecipitated using an anti-CRMP2 antibody. The samples were analyzed by SDS-PAGE, and immunoblotted with the indicated antibodies. CRMP2 was loaded as a control for protein input. **e** Lysates of the ovarian cancer cell line OVCA432, which was treated with non-targeting (nt) siRNA or *FER* siRNA, were immunoprecipitated with anti-CRMP2 antibody. The samples were analyzed by SDS-PAGE, and immunoblotted with the indicated antibodies. **f**, **g** The indicated recombinant CRMP2 wild-type or mutant proteins were purified and incubated with paclitaxel-stabilized rhodamine-labeled microtubules for 40 min at room temperature. Bar plots represent the mean + s.e.m. of microtubule width from at least 150 individual microtubules per condition tested. Shown are typical results from at least three independent replicates. Scale bar is 10 μm

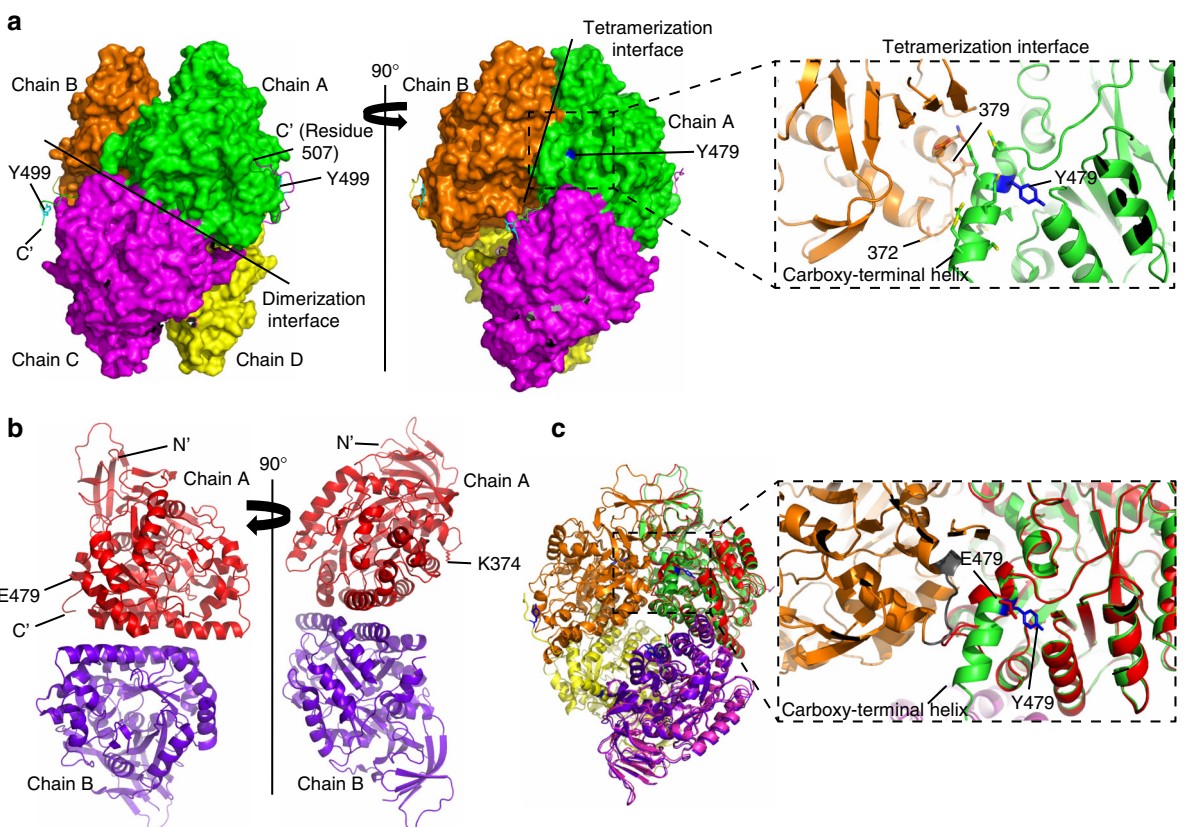

**Fig. 3** Phosphorylation of CRMP2 at Y479 induces oligomerization changes in CRMP2 structure. **a** (Left panel) Surface representation of a tetramer of CRMP2 (residues 13–516) showing the dimerization interface. Visible carboxy-terminal (C′) extensions of each chain are shown as ribbons. Reader-visible Y499 are labeled and shown in cyan. **a** (Right panel) Tetramerization interface between the dimers is shown. The position of Y479 at the interface is also highlighted in blue. Zoom-in shows the details of this interface comprising the carboxy-terminal helix of chain A and residues 372–379 of the neighboring chain B. **b** Crystal structure of the phosphomimetic mutant CRMP2–2E. (Left panel) Position of E479 is highlighted. (Right panel) Position of SUMOylation site, K374, is highlighted. **c** Superposition of CRMP2 wild-type tetramer structure and CRMP2–2E phosphomimetic mutant dimer structure. Zoom-in showing key differences at the tetramerization interface: chain A (green) of CRMP2 (residues 13–516) has the Y479 (blue) at the carboxy-terminal helix which interacts with the loop (gray) of chain B (orange). CRMP2–2E mutant (red) has E479 in the position. The mutation introduces a charge in the hydrophobic cavity, causing the carboxy-terminal helix to unwind and the tetramer to break

the tetramerization interface (Fig. 3a, right panel) shows that Y479 is positioned at the carboxy-terminal helix of each chain which is crucial for the dimers to associate and form tetramers. Tetramers are formed by the interaction of the carboxy-terminal helix from each dimer with residues 372–379 of the neighboring chain from the second dimer (Fig. 3a, right panel, zoom-in). Judging by the local environment around Y479, which is buried in a fairly hydrophobic cavity (Supplementary Fig. 5a), we estimate that if phosphorylated, it would cause a major change in the tetrameric state of the protein[24] and its microtubule bundling activity. Our wild-type CRMP2 protein construct has 26 extra residues (residues 491–516) at the carboxy terminus as compared to the constructs of the previously solved structures (PDB ID 2GSE and 5LXX). However, in the crystal structure, the electron density is only best visible up to residue 507, in 1 out of the 4 chains, suggesting flexibility in this region. The residues starting from 490 onwards seem to be interacting with their dimer partner either via hydrogen bonds or salt bridges (Fig. 3a, left panel). This interaction was not seen in previous structures because the constructs were too short. We speculate that the flexible carboxy terminus is required for the CRMP2 to probe and associate with the microtubules. The fact that Y499 is located in this region (Fig. 3a, left panel) and that our previous experiments with the carboxy-terminal truncation mutants showed this region to be important for bundling microtubules, we hypothesized that if

Y499 is phosphorylated, the carboxy terminus will not be able to bind the microtubules efficiently.

To test our hypotheses, we crystallized the phosphomimetic mutant of CRMP2 where the two tyrosine residues (Y479 and Y499) were mutated to glutamic acid (E). The structure (PDB ID 5MLE) confirmed that the mutant was a dimer instead of a tetramer (Fig. 3b). Comparison with wild-type CRMP2 showed that the phosphomimetic Y479E introduces a charge in a hydrophobic cavity that disrupts the tetramerization interface (Fig. 3c). Analytical gel filtration further validated that phosphomimetic mutant is a dimer in solution as compared to the tetrameric wild-type CRMP2 (Supplementary Fig. 5b).

We next tested the functional consequences of Y479 phosphorylation in the ovarian cancer cell line OVCA432 using ectopic expression of the wild-type CRMP2, CRMP2-Y479E phosphomimetic mutant, or CRMP2-Y479F phosphorylation-resistant mutant. As expected, the wild-type CRMP2 or its phosphorylation-resistant mutant co-localized with the metaphase spindle microtubules (Fig. 4a and Supplementary Fig. 6a). However, CRMP2-Y479E showed poor or no co-localization with metaphase spindle microtubules (Fig. 4a and Supplementary Fig. 6a). These results were consistent with our in vitro biochemical and structural data mentioned above. Taken together, our data suggest that the phosphorylation of CRMP2 at Y479 and/or Y499 impairs the microtubule

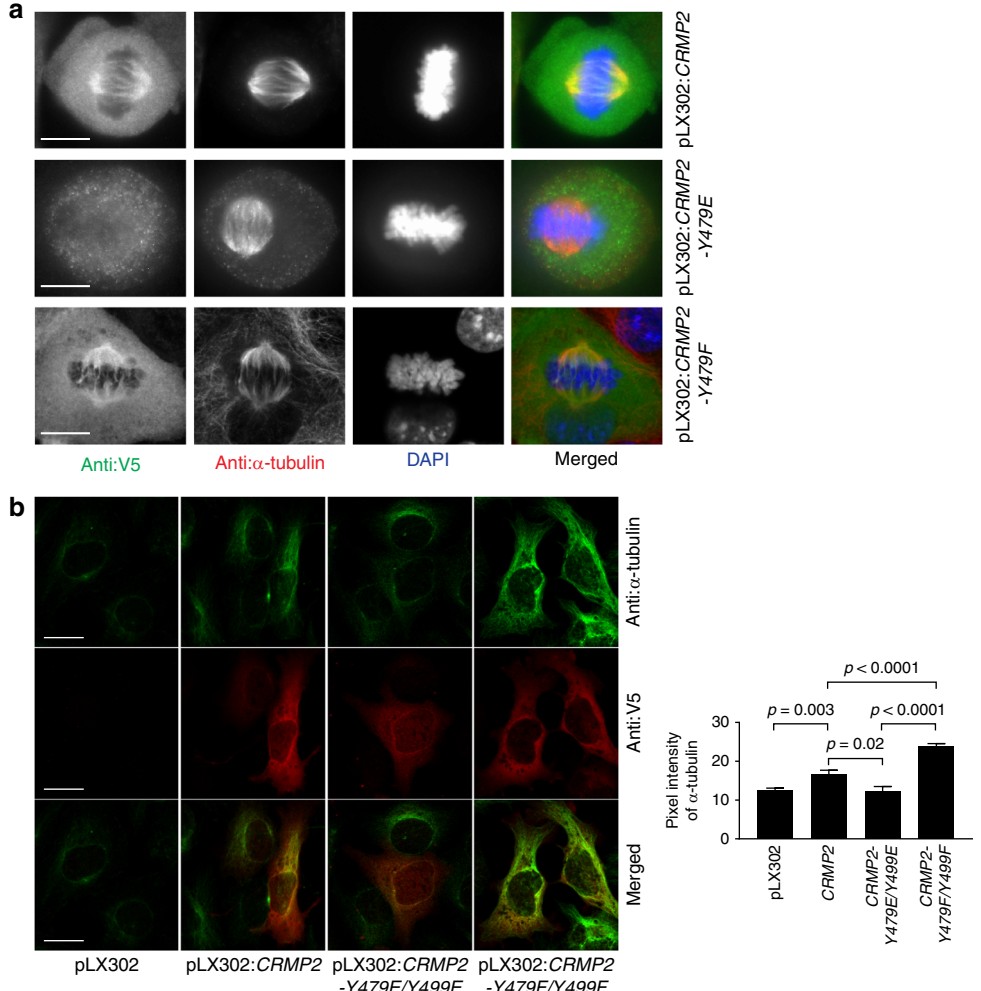

**Fig. 4** Y479 and Y499 play critical roles in CRMP2 microtubule function in ovarian cancer cells. **a** Immunofluorescence staining of OVCA432 mitotic cells which were transduced with pLX302:*CRMP2*, pLX302:*CRMP2-Y479E*, or pLX302:*CRMP2-Y479F* plasmids. CRMP2, CRMP2-Y479E, or CRMP2-Y479F were detected by anti-V5 antibody. Mitotic microtubules were detected using anti-α-tubulin antibody. The nucleus was revealed using DAPI. Scale bar is 10 μm. **b** Immunofluorescence staining of SKOV3 cells, which were transiently transduced with pLX302, pLX302:*CRMP2*, pLX302:*CRMP2-Y479E/Y499E*, or pLX302:*CRMP2-Y479F/Y499F* plasmid. At 48 h following transduction, the cells were incubated on ice for 25 min before fixation and staining using the indicated antibodies to reveal microtubules (anti-α-tubulin antibody) and CRMP2 (anti-V5 antibody). Bar plots represent the mean + s.e.m. of pixel fluorescence intensity values from at least 110 cells per condition tested, shown is a typical result from at least three independent replicates. Scale bar is 20 μm

bundling activity of CRMP2 because of impaired binding to microtubules.

**FER phosphorylation of CRMP2 inhibits its microtubule-stabilizing function in cells.** It is well known that microtubule bundling contributes to microtubule stability in cells. We therefore sought to determine the effect of CRMP2 phosphorylation on its microtubule-stabilizing activity in SKOV3 cells using cold-resistance assays. Ectopic expression of wild-type CRMP2 significantly increased resistance of microtubules to cold treatment compared to the empty vector control (Fig. 4b and Supplementary Fig. 6b), and this was consistent with our in vitro biochemical data (Fig. 1a). Moreover, ectopic expression of the phosphorylation-resistant double mutant CRMP2-Y479F/Y499F resulted in a further significant increase in resistance of microtubules to cold treatment compared to wild-type ectopic expression (Fig. 4b and Supplementary Fig. 6b). Taken together, our data demonstrate that CRMP2 Y479 and Y499 play critical roles in regulating the microtubule stabilization activity of CRMP2 in ovarian cancer cells.

**FER regulates microtubule stability via CRMP2 in ovarian cancer cells.** Given the strong biochemical evidence of how FER phosphorylation of CRMP2 regulates its microtubule-stabilizing function, we hypothesized that interfering with the function of FER may confer a therapeutic advantage by sensitizing cells to paclitaxel via the inhibition of CRMP2 phosphorylation. This was supported by the observation that FER was ubiquitously expressed in ovarian cancer cell lines (Fig. 5a, b) and strongly expressed in more than a third of 130 high-grade serous ovarian cancers (HGSOCs) (Supplementary Fig. 7a, b). In contrast, FES, the only other family member of FER, was undetectable at mRNA level in ovarian cancer cell lines (Supplementary Fig. 7c).

We first tested the effect of interfering with FER on microtubule length and stability. For the former, we utilized scratch assays to polarize microtubules perpendicular to the direction of the scratch as we previously described in SKOV3 cells[26], and measured the length of microtubules from the nuclear border to the leading edge of the cell. Inactivation of FER using the inhibitors TAE684 or WZ-4-49-8[22] in two independent ovarian cancer cell lines (SKOV3 and OVCA432) induced a

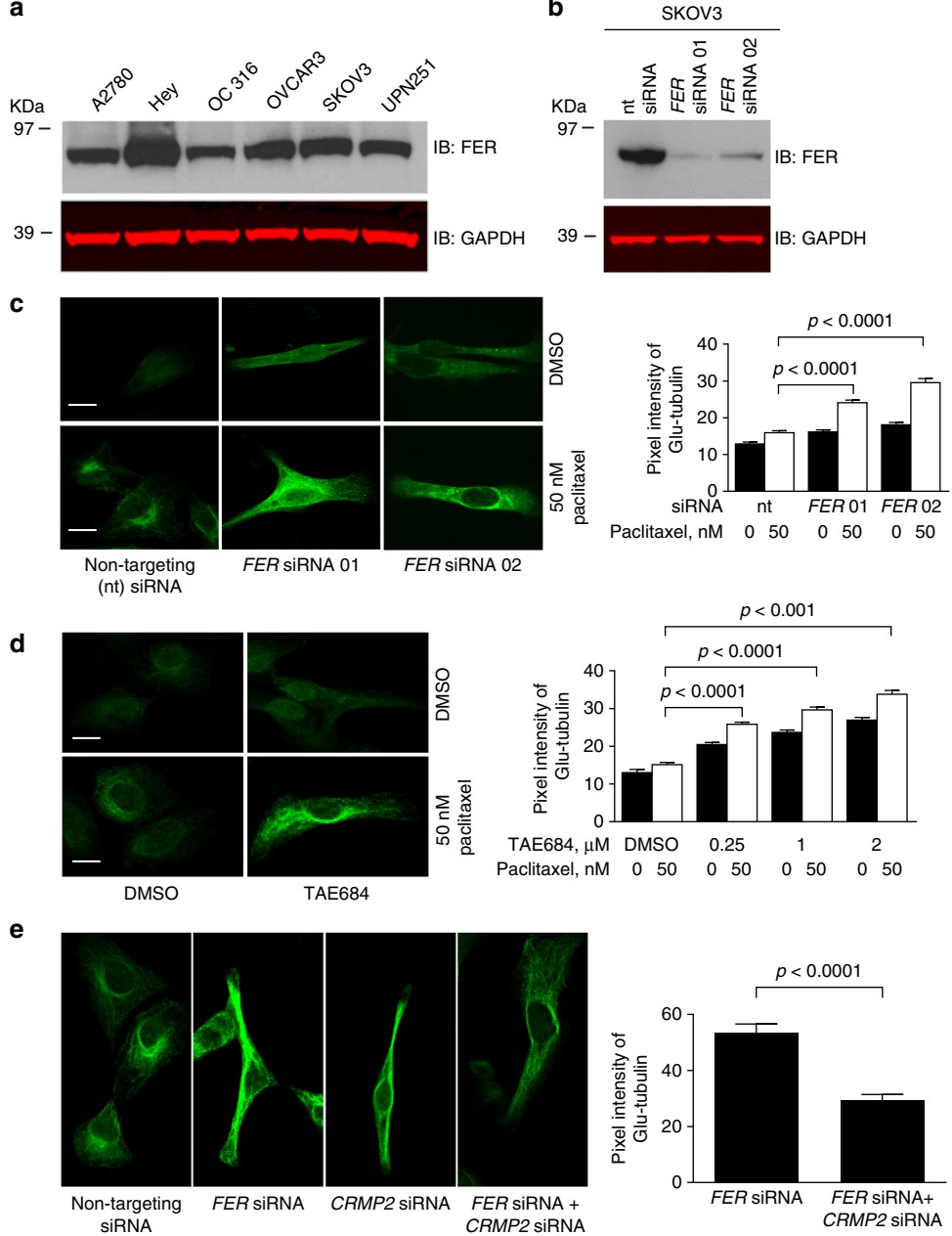

**Fig. 5** FER regulates microtubule stability through CRMP2 in ovarian cancer cells. **a** Western immunoblot (IB) analysis for FER expression in multiple ovarian cancer cell lines using the indicated antibodies. **b** Western blot analysis for FER expression in SKOV3 cells, which were transfected with non-targeting (nt) siRNA, *FER* siRNA 01, or *FER* siRNA 02. Total cell lysates were analyzed by SDS-PAGE and immunoblotted using the indicated antibodies. **c** Immunofluorescence staining for Glu-tubulin in SKOV3 cells, which were transfected with non-targeting (nt) siRNA, *FER* siRNA 01, or *FER* siRNA 02. Cells were treated with 50 nM paclitaxel or DMSO for 4 h before fixation. Bar plots represent the mean + s.e.m. of pixel fluorescence intensity values from at least 100 cells per condition tested, shown is a typical result from at least three independent replicates. Scale bar is 10 μm. **d** Immunofluorescence staining for Glu-tubulin in SKOV3 cells, which were treated with increasing concentrations of TAE684 overnight. Cells were treated with 50 nM paclitaxel or DMSO for 4 h before fixation and staining. Bar plots represent mean + s.e.m. of pixel fluorescence intensity values from at least 100 cells per condition tested, shown are typical results from at least three independent replicates. Scale bar is 10 μm. **e** Immunofluorescence staining for Glu-tubulin in SKOV3 cells, which were transfected with *FER* siRNA, *CRMP2* siRNA or both. Cells were treated with 50 nM paclitaxel or DMSO for 4 h before being fixed using 4% paraformaldehyde. Bar plots are shown as mean + s.e.m. of pixel fluorescence intensity values from at least 100 cells per condition tested, shown are typical results from at least three independent replicates. Scale bar is 10 μm

significant increase in microtubule length in a dose-dependent manner (Supplementary Fig. 7d). To further confirm this observation, we repeated the experiments using siRNAs depleting FER in SKOV3 and OVCA432 cancer cell lines. Similarly, depletion of FER increased microtubule length in both cancer cell lines (Supplementary Fig. 7e).

We also measured microtubule stability using an anti-detyrosinated tubulin (Glu-tubulin) antibody as we previously described[12,27]. Depletion of FER or its inhibition (using the TAE684 inhibitor) significantly increased microtubule stability in the SKOV3 ovarian cancer cell line in the presence or absence of low doses of paclitaxel treatment (Fig. 5c, d). Importantly, this

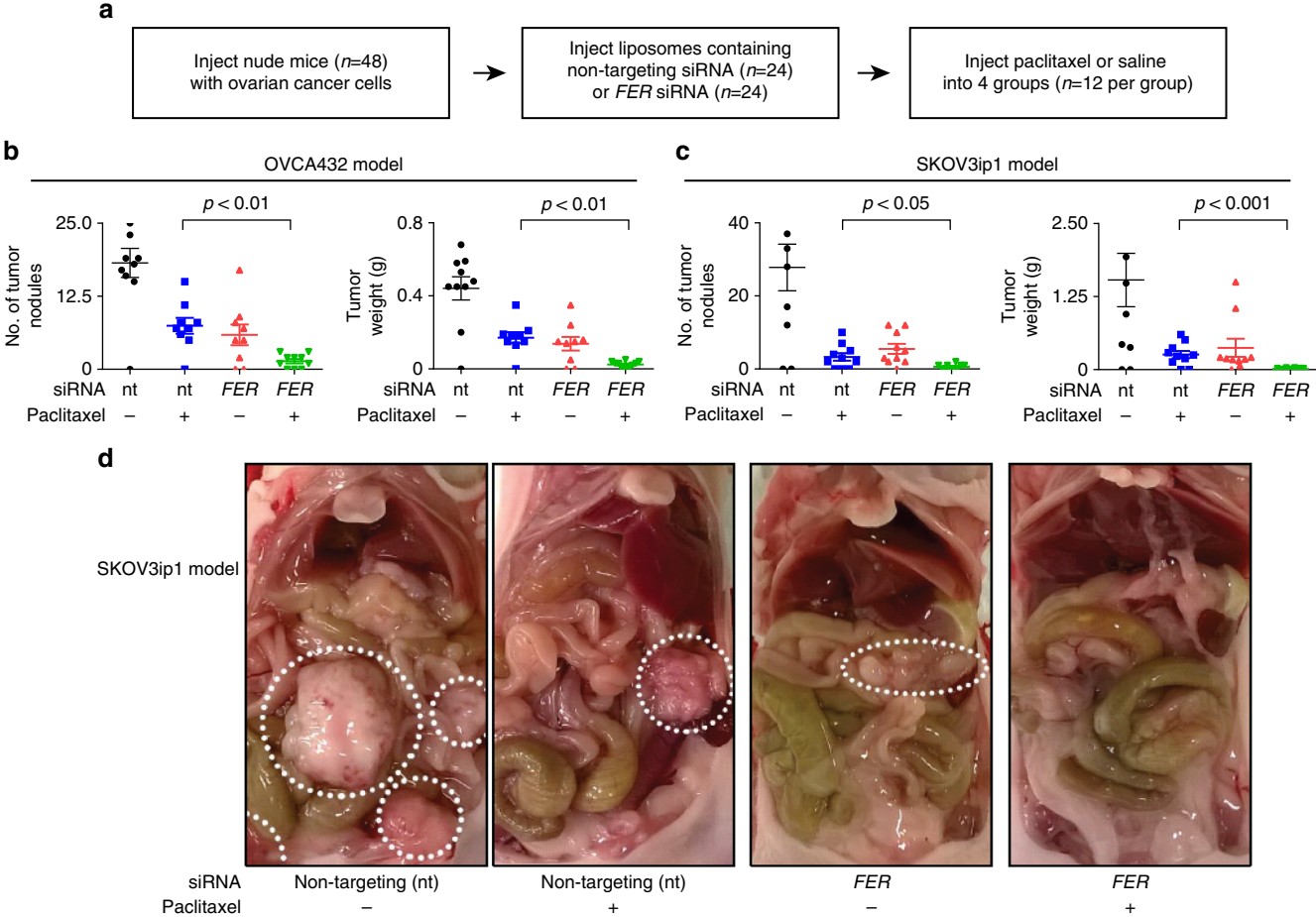

**Fig. 6** FER depletion enhances paclitaxel response in vivo. **a** Schematic representation of the design of the in vivo experiments. **b**, **c** Quantification analysis of the number of tumor nodules or tumor weight in the OVCA432 tumor cell model (**b**) and in the SKOV3ip1 tumor cell model (**c**). Bar plots represent the mean + s.e.m. for the number of tumor nodules and weight as indicated. **d** Representative images of tumor size for the SKOV3ip1 tumor cell model. The circles indicate the observed tumors

was achieved at low doses of the TAE684. This was further confirmed using a second FER inhibitor WZ-4-49-8 (Supplementary Fig. 7f). Importantly, depletion of CRMP2 rescued the increase in paclitaxel-induced microtubule stabilization following FER depletion (Fig. 5e). This result suggests that CRMP2 is required for enhancing microtubule stability following FER depletion. Additionally, CRMP2 was ubiquitously expressed in ovarian cancer cell lines (Supplementary Fig. 7g). It is interesting that both CRMP1 and CRMP4 were expressed at mRNA level in several ovarian cancer cell lines (Supplementary Fig. 7h). Furthermore, immunohistochemistry staining of more than130 HGSOCs showed that CRMP2 was strongly expressed in more than one third of ovarian cancers (Supplementary Fig. 7i, j). Since we found that FER and CRMP2 are involved in regulating microtubule stability, we tested whether FER regulates cell cycle progression. However, depletion of FER in SKOV3 cells had no effect on the cell cycle (Supplementary Fig. 7k).

**Down-regulation of FER sensitizes ovarian cancer cells to paclitaxel in culture and in vivo.** Since interfering with FER significantly increased microtubule stability (Fig. 5c, d) and since FER expression level correlates with paclitaxel half-maximal inhibitory concentration values in ovarian cancer cell lines (Supplementary Fig. 8a), we hypothesized that loss of FER may increase paclitaxel cytotoxicity in ovarian cancer cells. Depletion of FER using siRNAs significantly reduced cell proliferation

following paclitaxel treatment in multiple cell lines (Supplementary Fig. 8b–d). Furthermore, FER inhibition using TAE684, WZ-4-49-8, or TL-2-59 (compound 8)[28] in multiple cell lines (OC 316, SKOV3, OVCA432, A2780, and Hey) induced significant paclitaxel sensitization at a sub-stoichiometric paclitaxel concentration of 3 nM (Supplementary Fig. 8e–m). Critically, this sensitization was achieved at sub-lethal doses of the inhibitors that did not elicit a significant reduction in cell proliferation. Taken together, our data indicate that FER kinase is a potential therapeutic target for enhancing paclitaxel efficacy in ovarian cancer.

We next tested whether the down-regulation of FER might improve the therapeutic effect of paclitaxel in vivo. To test the combined effect of FER depletion and paclitaxel treatment, we employed two xenograft models of ovarian cancer (OVCA432 and SKOV3ip1 ovarian cancer cells) and performed in vivo FER depletion using the well-characterized neutral 1,2-dioleoyl-sn-glycero-3-phosphatidylcholine (DOPC) nano-liposomal system for delivery of siRNA as we previously described[26] (Fig. 6a). No significant difference was observed in the mouse body weight between the group of mice treated with paclitaxel alone and the group of mice treated with both FER siRNA and paclitaxel, in both of the two xenograft models of ovarian cancer (Supplementary Fig. 8n). However, in vivo depletion of FER (Supplementary Fig. 8o) induced a significant reduction in the number of tumor nodules and in tumor weight when combined with paclitaxel

treatment in comparison with paclitaxel treatment alone (Fig. 6b–d). Taken together, our data strongly suggest that the combination of FER depletion and paclitaxel treatment is an attractive therapeutic strategy in vivo to enhance paclitaxel response in ovarian cancer.

## Discussion

CRMP2 is one of the key modulators of microtubule dynamics but little is known about the signaling pathways that regulate its interaction with microtubules. In this work, we show how FER phosphorylates CRMP2 at Y479 and Y499 to regulate CRMP2 structure and its interaction with microtubules, respectively. We show that loss of CRMP2 phosphorylation, either via FER depletion or inhibition, significantly sensitizes ovarian cancer cells to paclitaxel treatment. Our data provide strong evidence that targeting FER is a plausible therapeutic strategy to enhance the efficacy of paclitaxel in treating ovarian and potentially other cancers. This is important because in spite of their wide use, taxanes only elicit a response in less than 50% of patients who receive it.

CRMP2 has previously been shown to associate with microtubules in cells, and to bind to tubulin heterodimers to promote microtubule assembly[13]. CRMP2 has been reported to be phosphorylated by GSK3β, CDK5, and Rho kinase at a number of serine and threonine residues at the carboxy terminus. These modifications impair the ability of CRMP2 to bind to tubulin[16,18,19]. FES, the only family member of FER, has previously been shown to phosphorylate CRMP2 at Y32[21,29]. However, our data demonstrate that phosphorylation mimicking at Y32 does not impair the microtubule bundling activity of CRMP2 (Supplementary Fig. 4). Our study reveals that FER, which was recently found to promote ovarian cancer metastasis[20], phosphorylates CRMP2 at two sites, Y479 and Y499. Both of these sites are important in regulating microtubule stability in vitro and in cells. A limitation of this study is that the anti-pY479-CRMP2 antibody was not suitable for immunohistochemistry. Therefore, it was not possible to test the correlation between the level of FER and pY479-CRMP2 in patient samples. It is, therefore, not possible to test whether or not the intrinsic FER activity can correlate with paclitaxel resistance in ovarian cancer patients.

CRMP2 is critical for axon formation by promoting microtubule assembly[13]. However, GSK3β and Rho kinase phosphorylate CRMP2 at T514 and S555, and inactivate it by impairing its association with tubulin dimers[18,19] and inhibiting microtubule assembly. Our study reveals that FER phosphorylated CRMP2 at Y479 and Y499, reducing its microtubule bundling activity (Fig. 2). Though all these three kinases have negative impact on the microtubule assembly activity of CRMP2, FER kinase regulates microtubule assembly via a different mechanism, compared with GSK3β and Rho kinase that are both known to be expressed in ovarian cancer cells[30,31]. Our study shows that phosphorylation at Y479 induces CRMP2 conformation change, impairing its tetramerization (Fig. 3), which is critical for its microtubule bundling function. However, how this conformation defect impairs CRMP2 activity should be investigated by resolving the structure of CRMP2 microtubule complex, using techniques such as cryo-electron microscopy (cryo-EM).

A previous report suggested that the YES kinase phosphorylates CRMP2 at Y479 and regulates T lymphocyte migration[23]. However, the biochemical and functional effect of such phosphorylation was not identified. Additionally, to our knowledge, this is the first report to show that FER phosphorylates Y499 of CRMP2. These two phosphorylation sites (Y479 and Y499) might act as biomarkers for detecting on-target activity of FER inhibitors in a therapeutic context, providing important translational significance. In the future, it will be interesting to investigate whether FER and YES co-regulate microtubule stability in ovarian cancers in terms of modulating CRMP2 activity on microtubules as YES is expressed in some ovarian cancer cell lines and also involved in the phosphorylation at Y479 of CRMP2 in the ovarian cancer cell line (Supplementary Fig. 3g–i). It will also be of interest to test whether targeting YES kinase can enhance paclitaxel response. It is important to note that CRMP1 and CRMP4 also are expressed in ovarian cancer cells (Supplementary Fig. 7h). Thus, it is possible that FER might also phosphorylate CRMP1 or CRMP4 at Y479 and Y499 in ovarian cancer cells since these two sites are highly conserved.

To understand the structural mechanism by which FER phosphorylation of CRMP2 affects its microtubule interaction, we solved structures of CRMP2 (wild-type and phosphomimetic mutant) (Fig. 3 and Supplementary Fig. 10). The new structural information has enabled us to speculate that both the intact tetramer conformation and the carboxy terminus of CRMP2 are pivotal for CRMP2 function to bundle microtubules.

Our data indicate that CRMP2 needs to be a tetramer to bundle microtubules effectively. Our wild-type structure depicts CRMP2 as a tetramer and accordingly, in in vitro microtubule bundling assay with CRMP2, the microtubules are seen as thick bundles. The non-mutated structure also shows that Y479 is at the tetramerization interface between two dimers. Once this interface is broken by phosphorylation of Y479, the tetramer would fall apart, as indicated by the structure of phosphomimetic mutant. This would cause a significant decline in microtubule bundling, which is what we observe in our in vitro microtubule bundling assay (Fig. 2f, g). Future work using cryo-EM will be important in visualizing the two modes of interaction. Our work reports the existence of CRMP2 as a dimer upon phosphorylation which, to our knowledge, has not been observed before. This is especially interesting, considering the recent study reporting an interplay between phosphorylation and SUMOylation regulating the biological functions of CRMP2[32]. The investigators show the existence of structurally conserved SUMOylation site, K374, which remains buried at the tetrameric interface of CRMP2. They further predicted that this site would be solvent accessible if interface was disrupted. Our phosphomimetic mutant structure, indeed, is in accordance with this prediction and clearly shows that K374 is solvent exposed and available for SUMOylation in the dimeric form (Fig. 3b, right panel).

Moreover, Y499 is located in the carboxy-terminal region of CRMP2. The carboxy terminus might be critical for binding microtubules, as shown in this and previous studies[14]. The importance of the carboxy-terminal region is further highlighted by the presence of post-translational modifications that are thought to be critical for regulating tubulin function[16,18,19]. These observations, along with the crystal structures that show poor electron density at the carboxy terminus, allow us to speculate that the carboxy-terminal region acts as a flexible anchor for interacting with microtubules and that phosphorylation reduces its ability to do so.

In summary, our work provides important insights into CRMP2-mediated bundling of microtubules and its regulation by FER phosphorylation. Due to the poor pharmacokinetics of paclitaxel, enhancing its efficacy at sub-stoichiometric concentrations is clinically important. We provide strong evidence that targeting FER may augment the therapeutic efficacy of paclitaxel at sub-stoichiometric concentrations by increasing microtubule stabilization.

## Methods

**Tissue culture cells**. A2780, Hey, OC 316, and UPN251 cell lines were obtained from Robert C. Bast lab in MD Anderson Cancer Centre. OVCA432 cell line was obtained from Anil K. Sood lab in MD Anderson Cancer Centre. OVCAR3, SKOV3, and HEK-293 cell lines were obtained from ATCC. SKOV3 were grown in

McCoy's 5A medium, supplemented with 10% fetal bovine serum (FBS) and 1% penicillin–streptomycin. The other cell lines were grown in Dulbecco's modified Eagle's medium supplemented with 10% FBS and 1% penicillin–streptomycin. The cell lines have been tested for mycoplasma contamination.

Validated non-targeting siRNA (UGGUUUACAUGUCGACUAA), two *FER* siRNAs (GGAGUGACCUGAAGAAUUC and GGAAAGUACUGUCCAAAUG), and *CRMP2* siRNA (GAAGGGAACUGUGGUGUAU) were purchased from Dharmacon, GE Healthcare. OVCA432 and SKOV3 cells were transfected with 40 nM siRNA using Lipofectamine RNAiMax (Thermo Fisher Scientific) according to the manufacturer's instructions.

For transient transfections, plasmid DNA and FuGENE HD reagent (Promega) were diluted in Opti-MEM medium in a 1:3 ratio at 3.0 μg DNA per well, and incubated for 15 min at room temperature before being added to plated cells.

OVCA432 cells were used for lentiviral transduction. Lentiviral vectors were packaged and infected as following description. In brief, the envelop plasmid pMD2.G (a gift from Dr. Didier Trono; Addgene, 12259), packaging plasmid psPAX2 (a gift from Dr. Didier Trono; Addgene, 12260), and pLX302 (a gift from Dr David Root[33]) based vectors were cotransfected into HEK-293, and viruses were harvested 48 h after transfection. Then, OVCA432 cells were transduced with lentiviral particles carrying plasmids containing the gene of interest. Primers used for lentiviral constructs are listed in Supplementary Table 1. Puromycin (0.5 mg/ml) selected colonies were screened for further analysis.

**Cloning and site-directed mutagenesis.** CRMP2 complementary DNA (cDNA) was obtained from OriGene. Recombinant CRMP2 DNA was sub-cloned into individual vector after DNA sequence verification. Recombinant DNA was amplified by specific primers listed in Supplementary Table 1.

**Reverse and real-time qPCR.** Total RNA was isolated from cell lines using RNeasy mini Kit (Qiagen) and cDNA was synthesized using TaqMan Reverse Transcription kit (Applied Biosystems). Quantitative (real-time) PCR (qPCR) was conducted in a 20 μl reaction volume using the SYBR Green PCR Master Mix (Applied Biosystems) according to the manufacturer's introductions. The samples were analyzed on an ABI Prism 7000 sequence detection system (Applied Biosystem, Foster City, USA). The primer sequences for qPCR are provided in Supplementary Table 2.

**Recombinant protein production.** Protein expression constructs were transformed into *Escherichia coli* Tuner (DE3) strains to generate the recombinant proteins, which were induced by 0.5 mM isopropyl β-D-thiogalactopyranoside and incubated at 16 °C overnight. Bacterial cells were harvested at 4000×*g* and lysed by sonication in the presence of the buffer containing 500 mM NaCl, 10% glycerol, 50 mM HEPES, pH 7.5, and 0.5 mM Tris(2-carboxyethyl)phosphine hydrochloride (TCEP). Further purification of these proteins was performed with nickel-nitriloacetic acid resin (Novagen) according to the manufacturer's instructions. Proteins were dialyzed against a buffer containing 300 mM NaCl, 10% glycerol, 20 mM HEPES, pH 7.5, and 0.5 mM TCEP. Further purification of the proteins was done by size exclusion gel filtration in 300 mM NaCl, 10% glycerol, 20 mM HEPES, pH 7.5, and 0.5 mM TCEP. The proteins were concentrated and snap frozen in liquid nitrogen before storage at −80 °C.

**In vitro kinase assay.** Commercial FER proteins were obtained from Thermo Fisher Scientific. Then, 12 μM CRMP2 wild-type or CRMP2 mutant proteins and 3 μM FER protein were incubated with 80 μM ATP (SignalChem) in kinase assay buffer 1 (SignalChem, 25 mM MOPS, pH 7.2, 25 mM MgCl₂, 5 mM EGTA, 2 mM EDTA, 0.25 mM dithiothreitol (DTT), 12.5 mM β-glycerol-phosphate) in a 25 μl reaction. Reactions were incubated at 30 °C for 20 min before adding 2× NuPAGE LDS sample buffer (Thermo Fisher Scientific) and boiling at 95 °C for 5 min.

**Mass spectroscopy.** LC-MS/MS experiments were performed using a Dionex Ultimate 3000 RSLC nanoUPLC (Thermo Fisher Scientific Inc., Waltham, MA, USA) system and a QExactive Orbitrap mass spectrometer (Thermo Fisher Scientific Inc.). Separation of peptides was performed by reverse-phase chromatography at a flow rate of 300 nl/min and a Thermo Scientific reverse-phase nano Easy-spray column (Thermo Scientific PepMap C18, 2 μm particle size, 100 A pore size, 75 μm i.d. x 50 cm length). Peptides were loaded onto a pre-column (Thermo Scientifc Pepmap 100 C18, 5 μm particle size, 100 A pore size, 300 μm i.d. x 5 mm length). The LC eluent was sprayed into the mass spectrometer by means of an Easy-spray source (Thermo Fisher Scientific Inc.). All *m/z* values of eluting ions were measured in an Orbitrap mass analyzer, set at a resolution of 70,000. Data-dependent scans (Top 20) were employed to automatically isolate and generate fragment ions by higher energy collisional dissociation in the quadrupole mass analyser and measurement of the resulting fragment ions was performed in the Orbitrap analyser, set at a resolution of 17,500. Peptide ions with charge states of between 2+ and 5+ were selected for fragmentation. Data were processed using Protein Discoverer (version 1.4., Thermo Fisher). Briefly, all data were converted to mgf files and submitted to the Mascot search algorithm (Matrix Science, London, UK) and searched against a custom database containing only the CRMP2 sequence

using a modification of tyrosine phosphorylation (Y). Peptide identifications were accepted if they could be established at greater than 95% probability.

**Western blotting.** Samples were separated by sodium dodecyl sulfate–polyacrylamide gel electrophoresis (SDS-PAGE) using NuPage Bis-tris polyacrylamide gels (Thermo Fisher Scientific) and transferred to nitrocellulose membrane using the iBlot dry gel transfer system (Invitrogen). Membranes were incubated with primary antibodies overnight at 4 °C before washing with Tris-buffered saline and 0.1% Tween-20. Secondary antibodies (anti-rabbit horseradish peroxidase (HRP) conjugated and anti-mouse HRP conjugated, Abcam) were incubated with membranes for 1 h at room temperature. Rabbit anti-CRMP2-pY479 and rabbit anti-CRMP2-pY499 antibodies were generated by Biogene, using peptides (CFPDFV-pY-KRIKA, for pY479) and (CVPRGL-pY-DGPV, for pY499) to immunizing rabbits. They were used in 1:1000 dilutions for western blot. The information and used dilutions of commercial primary antibodies used in this paper is provided in Supplementary Table 3. The original western blots are provided in Supplementary Fig. 9.

**Immunoprecipitation.** Cells were seeded into 10 cm culture dishes before lysis in lysis buffer (50 mM HEPES, pH 7.0, 1.5 mM MgCl₂, 1 mM EGTA, 100 mM NaF, 10 mM sodium pyrophosphate, 150 mM NaCl, 10% glycerol, 1% Triton X-100, 1 mM Na₃VO₄, 10 μg/μl Aprotinin, 10 μg/ml Leupeptin, and 1 mM phenylmethylsulfonyl fluoride). Lysates were pre-cleared with 40 μl protein G sepharose slurry for 30 min. Then, 2 mg cell lysate was incubated with 3 μg antibody overnight at 4 °C, followed by 60 μl protein G sepharose slurry incubation for 4 h at 4 °C. Samples were washed comprehensively with lysis buffer five times prior to addition of 20 μl 4× NuPAGE sample buffer (Thermo Fisher Scientific). Western blots were performed as described previously.

**Immunofluorescence and confocal microscopy.** Cells were grown on glass coverslips prior to fixation with 4% paraformaldehyde for 3 min at room temperature and subsequent permeabilization with ice-cold 100% ethanol overnight at −20 °C. Cells were then washed three times with wash buffer (Tris-buffered saline, 0.2% Triton X-100 (Sigma) and 0.04% SDS (Sigma)). Cells were further incubated with blocking buffer (Tris-buffered saline, 0.2% Triton X-100 (Sigma), 0.04% SDS (Sigma) and 1.5% bovine serum albumin (Sigma)) for 30 min and incubated with primary antibodies for 1 h at room temperature. Cells were incubated with fluorescent secondary antibodies (Alexa fluor-488 or Alexa fluor-568, Invitrogen) for 1 h at room temperature in the dark. Images were captured using the Zeiss microscopy with a 63 ×oil objective.

**Microtubule polymerization and bundling visualization.** Tubulin was prepared from pig brains according to the previous report[34]. Then, 30 μM tubulin in PEM buffer (80 mM PIPES-KOH, pH 6.9, 1 mM EGTA, and 1 mM MgCl₂) were used to investigate microtubule polymerization in the absence or presence of CRMP2 or pCRMP2 proteins at the indicated concentration. Microtubule polymerization was analyzed by measuring the absorbance at 350 nm at 37 °C. This assay was performed using a SpectraMax M2e microplate reader (Molecular Devices).

Rhodamine-tubulin was labeled with 5-Carboxy-tetramethylrhodamine *N*-succinimidyl ester (Sigma-Aldrich), according to the previous report[35]. Rhodamine microtubules were incubated with PEM buffer (80 mM PIPES-KOH, pH 6.9, 1 mM EGTA, and 1 mM MgCl₂) in the presence of 10 μM paclitaxel in a 37 °C water bath for 40 min. Rhodamine-labeled microtubules were incubated with recombinant CRMP2 at the ratio of 4:1 for 30 min at room temperature. Then, 2 μl of each reaction was mounted on a poly-lysine-coated coverslip and samples were observed with the Zeiss microscopy using a 100× oil immersion objective. For microtubule bundle width calculation, we measured the maximum width of each microtubule, using Image J software. More than 150 microtubules (more than 8 μm of length) were randomly selected to be measured.

Microtubule bundling was also observed via negative stain electron microscopy. A total of 10 μl pre-assembled microtubules in the absence or presence of CRMP2 were incubated with fresh glow discharge carbon 300 mesh copper grids (TAAB laboratories) for 2 min. Grids were then blotted with filter paper before being stained with 2% uranyl acetate for 10 s. Grids were imaged on a FEI Tecnai 12 TEM equipped with a Gatan US1000 CCD camera at 120 kV before being air dried.

**Microtubule elongation visualization by TIRFM.** Flow cells were prepared as previously described[36]. Flow cells were incubated with 1 μg/ml anti-α-tubulin antibody (Sigma-Aldrich) for 5 min, followed by washing with PEM buffer and 1% Pluronic F127 (Sigma-Aldrich). Next, 400 nM rhodamine-GMPCPP (Guanosine-5′-[(α,β)-methyleno]triphosphate) microtubules (10% rhodamine labeled) were injected into flow cell after it was washed with PEM buffer again. GMPCPP (Jena Bioscience)-labeled microtubules were prepared as previously described[37]. After 5 min, the flow cell was washed with 1 volume of PEM buffer, and then was injected with 7.5 μM rhodamine-tubulin (50% labeled) in the presence of 400 nM CRMP2 or pCRMP2. The reaction was conducted in the following buffer: 80 mM PIPES-KOH, pH 6.9, 1 mM EGTA, 1 mM MgCl₂, 50 mM NaCl, 50 mM DTT, 1 mM GTP, 15 mM glucose, 20 mg/ml catalase, 100 mg/ml Glucose oxidase, and 0.5% methylcellulose. Single microtubule filaments were observed immediately by TIRF

illumination with an Olympus IX83 microscopy equipped with a 150× oil objective (1.45 numerical aperture). Images were collected for 5 min with an interval of 5 s using a Photometrics Evolve Delta EMCCD camera (Photometrics) with Olympus cellSens software. For microtubule elongation rate calculation, we measured the length of a single microtubule at starting point (T1, second), which gave the length (L1, μm). Then, we selected the ending point (T2, second), and measured the length of the same single microtubule (L2, μm). The length of microtubules was measured using Image J software. More than 25 microtubules were randomly selected to be measured. Microtubule elongation rate was calculated using the following equation: $(L2-L1)/[(T2-T1)\times 60]$.

**Colony formation assay**. A2780, Hey, OC316, OVCA432, and SKOV3 cells were seeded into 6-well plates at 500 cells/well. After 48 h, cells were treated with indicated concentrations of FER inhibitors (TAE684, WZ-4-49-8, or TL-2-59) or dimethyl sulfoxide (DMSO) vehicle control as indicated in the figure legend (Supplementary Fig. 8e–m). After 48 h of inhibitor treatment, cells were treated with or without 3 nM paclitaxel in the presence of FER inhibitors. After 6 days, plates were fixed with 50% methanol and stained with Coomassie Brilliant Blue (Sigma-Aldrich).

**Cell cycle analysis**. Analysis of SKOV3 cell cycle was performed using flow cytometry and 4',6-diamidino-2-phenylindole (DAPI) staining. SKOV3 cells were trypsinized and re-suspended in 1.2 ml phosphate-buffered saline (PBS). Then, 3 ml ice-cold 95% ethanol was added dropwise while being vortexed, following by two washes with PBS. Next, $1 \times 10^6$ cells were re-suspended in 1 ml DAPI (D9542) staining solution, which contains 0.1% Triton X, 1 μg/ml DAPI, and PBS. Cells were applied to analysis after incubation on ice for 30 min.

**Immunohistochemistry**. Samples for tissue arrays were collected retrospectively within a translational study that was ethically approved (Number 11/SC/014, Berkshire NRES Committee).

To investigate the expression level of FER or CRMP2 in high-grade serous ovarian cancers, tissue arrays comprising samples from 130 HGSOCs were used. Tissue sections of 4 μm thickness were cut from the samples. Automated staining was performed with the Leica BOND-MAX autostainer (Leica, Microsystem) using the following conditions: (1) antigen retrieval at 100 °C for 20 min with Epitope Retrieval Solution 1 (AR9961, Leica Biosystems); (2) primary antibody incubation with the FER antibody (ab130199) at 1:300 dilution or the CRMP2 antibody (ab129082) at 1:200 dilution for 15 min; (3) then detection using the BOND™ Polymer Refine Detection System (DS9800, Leica Biosystems) as per the manufacturer's instructions. Stained slides were scanned at 40× magnification using the Aperio slide scanner (Aperio). The ImageScope software (v11.2.0.780, Aperio) was used for quantification of cytoplasmic staining. To score FER or CRMP2 positivity in the samples, regions of interest (ROIs) were defined, using the "positive pen tool". The marked ROIs were analyzed with the algorithm "positive pixel count v9". It scored the intensity of staining of all cells within the marked areas and automatically classified cells into 4 categories according to the pixel intensity values: (1) no detectable signal, (2) weak intensity of staining, (3) moderate intensity of staining, and (4) strong intensity of staining. Sum of (1), (2), and (3) was defined as low expression, and (4) was defined as high expression. The percentage of cells with low or high expression of FER or CRMP2 to the total number of cells scored in a ROI was presented.

**In vivo experiments**. Nude immunodeficient C57BL/6 mice were obtained from the National Cancer Institute, Frederick Cancer Research and Development Center (Frederick, MD, USA), and were bred and maintained according to institutional guidelines. All animal experiments were performed in accordance with the protocols approved by the MD Anderson Cancer Center Institutional Animal Care and Use Committee.

To investigate the combinatorial effects of paclitaxel treatment and FER depletion on tumor growth, the ovarian cancer cell lines SKOV3ip1 or OVCA432 were injected into the peritoneal cavity of female nude mice at approximately 12 weeks of age. To induce tumor development, SKOV3ip1 or luciferase-labeled OVCA432 ovarian cancer cells were trypsinized and suspended in 200 μl of Hanks' balanced salt solution (Gibco, Carlsbad, CA) before being injected into mice intraperitoneally (SKOV3ip1, $1.0 \times 10^6$ cells/mouse; OVCA432, $1.5 \times 10^6$ cells/mouse). After 1 week, four separate groups were treated as follows: (1) control non-targeting siRNA/DOPC; (2) non-targeting siRNA/DOPC + paclitaxel; (3) FER siRNA/DOPC; and (4) combination of paclitaxel and FER siRNA/DOPC. Each treatment was administered intraperitoneally. Each group consisted of 10 mice. Paclitaxel was introduced 1 day after siRNA incorporation for OVCA432 at a dose of 35 μg per mouse once a week. At 4 weeks after treatment (or mice in any group became moribund), all four groups of mice were sacrificed and the mouse weight, tumor weight, and number of tumor nodules were recorded. In addition, tumor tissue was fixed in formalin for paraffin embedding.

siRNA for in vivo delivery was incorporated into DOPC (Avanti Polar Lipids, Inc.) as described previously[38,39]. Briefly, 5 μg of siRNA and DOPC were mixed in the presence of excess tertiary butanol at a ratio of 1:10 (w/w). After addition of Tween-20 (1:19 Tween-20:siRNA/DOPC), the mixture was frozen in an acetone dry ice bath and lyophilized. Liposomes were reconstituted with 200 μl (to achieve a 200 μg/kg concentration) PBS and administered into mice intraperitoneally. Mice were treated for 6 weeks biweekly to have siRNA bioavailability and sustained silencing effect.

**CD spectrophotometry**. CD spectra were recorded on a J-815 spectropolarimeter (JASCO) at 20 °C with a scan speed of 100 nm/min using 0.1 cm path length quartz cells from Starna Scientific UK. The concentration of both full-length CRMP2 wild-type and full-length CRMP2–6F was 0.2 mg/ml. Data points were collected with a resolution of 0.2 nm, an integration time of 1 s, and a slit width of 1 nm. Each spectrum shown is the result of nine averaged consecutive scans from which buffer scans were subtracted.

**Analytical gel filtration**. CRMP2 wild type (residues 13–516) and CRMP2–2E (residues 13–516) at concentrations of 0.5 mg/ml were run on Sepax SRT SEC-300 column connected to Dionex™ HPLC System in buffer containing 300 mM NaCl, 10% glycerol, 20 mM HEPES, pH 7.5, and 0.5 mM TCEP with injection volume of 65 μl.

**Structure determination**. CRMP2 (residues 13–516) was crystallized using sitting drop vapor diffusion method at 20 °C by mixing 100 nl of protein (19 mg/ml) with 50 nl of crystallization solution (0.2 M ammonium acetate, 30% PEG 4000, and 0.1 M citrate buffer, pH 5.5) and equilibrating against 20 μl of crystallization solution. Crystals were cryo-protected with 25% ethylene glycol mixed in crystallization buffer. X-ray diffraction dataset was collected to 1.8 Å at 100 K on Diamond synchrotron beamline I03. The data were processed using XDS and 5% of the reflections were randomly assigned to calculate the free R factors. The structure was phased using CCP4- Phaser[40] by providing CRMP2 tetramer (residues 13–490, PDB: 2GSE)[25] as the template. Single copy of the tetramer was found in the asymmetric unit. The structure was refined using CCP4-Refmac5[41]. The data collection and refinement statistics are provided in Supplementary Table 4.

CRMP2 (residues 13–516)-2E phosphomimetic mutant was crystallized using sitting drop vapor diffusion method at 20 °C by mixing 50 nl of protein (12.91 mg/ml) with 100 nl of crystallization solution (0.2 M magnesium chloride, 25% PEG 3350, and 0.1 M bis-tris buffer pH 6.5) and equilibrating against 20 μl of crystallization solution. Crystals were cryo-protected with 25% ethylene glycol mixed in crystallization buffer. X-ray diffraction dataset was collected to 2.48 Å at 100 K on Diamond synchrotron beamline I04–1. The data were auto-processed using xia2[42]. The structure was phased using CCP4- Phaser[40] by providing a monomer from CRMP2 tetramer (residues 13–490, PDB: 2GSE)[25] as the template. Two copies of the monomer were found in the asymmetric unit. The structure was refined using CCP4-Refmac5[41]. The data collection and refinement statistics are provided in Supplementary Table 4.

**Quantification and statistical analysis**. Levels of significance were performed with Student's t-test while comparing two groups or with one-way analysis of variance while comparing more than two groups using GraphPad Prism, and means+s.e.m. values were presented.

**Data availability**. The PDB ids for CRMP2 (residues 13–516) and CRMP2–2E (residues 13–516) structures reported in this paper are: 5MKV and 5MLE, respectively. All other remaining data are available within the article and Supplementary Files, or available from the authors upon request.

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

## Acknowledgements

We thank the Oxford Centre for Histopathology Research for their help with processing tissue arrays. We thank Errin Johnson, Anna Pielach, and the Dunn School Bioimaging Facility for technical help with the electron microscopy work. We thank Christoffer Lagerholm and Esther Garcia in Wolfson Imaging Centre Oxford for technical help in total internal reflection fluorescence microscopy (TIRFM) work. We thank Dr. David Staunton at the Department of Biochemistry, University of Oxford, for his technical help and assistance in performing and analyzing circular dichroism spectrophotometry experiments. This work was funded by Target Ovarian Cancer, the Medical Research Council, the National Institute for Health Research (NIHR) Oxford Biomedical Research Centre (BRC), Experimental Cancer Medicine Centre, and the Helen Clarke Fund.

## Author contributions

A.A.A. and Y.Z. conceived the project. Y.Z., R.S., F.v.D., and A.A.A. wrote the manuscript. Y.Z., R.S., L.S.M., C.T., J.G., M.W., K.M., E.M.C., D.M., F.M., S.H.-G., K.H., F.C., A. Alsaadi, A. Albukhari, D.C.F., D.J., S.P., C.R.-A., G.L.-B., L.C., and D.F. conducted experiments. A.A.A., C.Y., S.K., N.S.G., K.M., S.D., R.C.B., F.v.D., and A.K.S. supervised research.

## Additional information

**Competing interests:** The authors declare no competing financial interests.

