## [Peer review file · Nature Communications]

Reviewers' comments:

Reviewer #1 (Remarks to the Author):

This is a well executed study addressing the issue of controlling the state of microtubule dynamics in order to enhance the tumor responses to paclitaxel. Specifically the authors convincingly demonstrate that phosphorylation of the microtubule associated protein, CRMP2, by FER kinase, reduces its microtubule polymerizing activity, and its ability to induce microtubule elongation and to bundle microtubules. The authors identified by phosphopeptide mapping six phosphorylated sites among which two displayed structural and functional roles in this context. By solving the crystal structure of the wild type and the phosphomimetic mutants they propose a mechanism through which the phosphorylation impairs the tetramer conformation of CRMP2 thus impairing binding to microtubules. Notably one of this phosphorylation sites has been identified in a previous publication. The specific contribution of FER to this biological process is further demonstrated in this work by measuring the outcome of its inhibition and /or its reduced expression by siRNA, on the cytotoxic responses to paclitaxel. This was done in several cell lines and in xenograft mouse models of ovarian cancer. Altogether this is an interesting study which combines biochemical assays, protein structural studies, in vitro microtubule bundling assays, with cell based responses to paclitaxel, and might have a potential clinical implication in cancer.

Comments and suggestions:

1. As mentioned by the authors, several Ser/Thr key kinases were previously shown to phosphorylate the C-terminal part of CRMP2 at other sites, which also block its ability to bind to tubulin. For example, Thr 514 phosphorylation by GSK-3 β . The authors should refer to these other functional phospho sites in the Discussion, and explain the difference/ common features concerning the outcome of these multiple kinases.
2. It is interesting that FER shares with FES kinase the phosphorylation on Y32. The importance of this site was excluded in this work since its substitution to phenyl alanine in the context of CRMP2-4F construct retained bundling activity and remained sensitive to FER. Yet the authors should test the possibility that a phospho mimicking mutation at this site might have an effect on its own or might synergize with the other two sites under some conditions.
3. Is the Yes kinase expressed in the ovarian cell lines tested in this work? If positive what is the outcome of Yes knock down both at the cellular level and on Y479 phosphorylation? This should be examined. What is the range of specificity of the FER inhibitors used in his work?
4. While the suppression of FER activity/expression affects the cytotoxic responses to paclitaxel in several ovarian cell lines, a statistically significant correlation between tumor responses to paclitaxel in patients and FER expression levels and/or Y479 phosphorylation levels is still missing. Thus, the reason for tumor resistance to paclitaxel is not clear yet and can not be attributed exclusively to FER levels at this stage. This should be discussed.

Reviewer #2 (Remarks to the Author):

Major:

1. The authors should include relevant literature supporting FER kinase as an oncogene. The currently cited reference (12) did not specifically describe FER as an oncogene.
2. If there is a lack of literature supporting FER expression in human ovarian malignancies, a detailed, large-scale expression study in ovarian tumour tissues and normal tissues should be performed. Association of expression with pathological or clinical parameters as well as the relationship between FER and/or CRMP2 expression and patient clinical outcome should be conducted.

3. Along the same lines, since an antibody against FER-phosphorylated-CRMP2 at the Y497 amino acid residue has been generated and the antibody specificity was assessed (e.g. Fig. 2e), it would be informative to assess the correlation between FER expression level and pY497 levels of CRMP2 on a panel of human tumor tissues and cell line models. Additionally, it would be important to use both in vitro and in vivo models to demonstrate whether CRMP2 pY497 levels can be inhibited by FER kinase inhibitor and, if yes, to what extent as compared to the FER knockdown approach.

The goals of the above experiments are to provide additional supports for the presence and prevalence FER/CRMP2 regulatory axis in in vivo context.

4. To determine the therapeutic potential of the EFR targeting strategy, downregulation of EFR kinase was performed using siRNA or EFR inhibitors (e.g., TAE684, WZ-4-49-8) (Page 10, Fig. 5cd and Fig. 6); however, whether the approaches affect FER kinase activity was not presented or described. Since this is an important assessment to evaluate the specificity of the targeting strategy, such experiments should be performed.

5. Because paclitaxel targets microtubules, their results suggested that the FER/CRMP2 axis may also regulate microtubule function, which is essential for mitosis. Can the authors detect alterations of the cell cycle and apoptosis when they modulate the FER/CRMP2 axis?

6. Figure 4b: Did pLX302:CRMP2-Y479E/Y499E show any significant change pertaining to resistance of microtubules to cold treatment compared with wild type CRMP2 and CRMP2-Y479F/Y499F? The authors did not mention this in the Results section.

7. Fig. 5 and Supplementary Fig. 5: Does the endogenous expression of FER, CRMP2, or phosphorylated CRMP2 correlate with sensitivity to paclitaxel in this panel of ovarian cancer cells?

8. Methods section on page 30 – Was the tumor xenograft study performed on C57BL/6 mice or on immunodeficient mice?

9. The second paragraph on page 9 (see below) is confusing. Was FER expressed in ovarian cancer cell lines or was un-detectable in ovarian cancer cell lines?

“This was supported by the observation that FER was ubiquitously expressed in ovarian cancer cell lines (Fig. 5a) and strongly expressed in more than a third of 130 high grade serous ovarian cancers (HGSOCs) (Supplementary Fig. 5a,b). In contrast, FER was undetectable at protein (data not shown) or mRNA level in ovarian cancer cell lines (Supplementary Fig. 5c).”

10. There are some issues pertaining to the animal study (Fig. 6 and supplementary Fig 6). To help evaluating the specificity of the observed in vivo phenotypes, the following experiments should be performed:

(1) The knockdown efficiency of these nanoliposomes should be described. In addition, the distribution of DOPC-nanoliposomes in mouse body and whether they selectively target tumour tissues should be demonstrated. Furthermore, their effects on the mouse body weight should be measured.

(2) Detailed information about the preparation, dosage, and treatment cycles of DOPC-nanoliposomes should be provided in Materials and Methods.

(3) Since an ovarian cancer cell line with engineered expression of luciferase reporter was employed in the animal study, tumor growth kinetics measured by in vivo bioluminescent images should be demonstrated in this report.

.

(4) If FER inhibitor is proven to be safe for clinical use, it can be included as another agent for a proof-of-principal study.

11. Quantification and statistical analysis - many of the studies are comparing more than two group means; t-test is inappropriate to be used. The authors should consider using one-way ANOVA when comparing means of more than two groups.

Minor:

- Please specify how microtubule elongation and microtubule width were measured and calculated.
- Based on the description, it is not clear how many different FER inhibitors were used for this study. The authors sometimes described TAE684 and WZ-4-49-8, and at other times described TAE684 and TL-2-59.
- Is the image shown in Fig 5C generated from FER 01 siRNA or FER 02 siRNA? Please include images for both siRNAs.
- In terms of measuring microtubule dynamics in live cells, the authors should consider tracking fluorescently labelled plus-end-tracking proteins over time, performing fluorescence recovery after photobleaching (FRAP) on cells transfected with plasmid encoding fluorescently labelled proteins, or method mentioned in the following paper: PMID 21773917.

Reviewer #3 (Remarks to the Author):

Zheng et al. "Tuning microtubule dynamics..."

The paper discusses the interplay of CRMP2 phosphorylation by FER kinase, CRMP2 structure, and microtubule dynamics, in order to pave new ways for cancer therapeutics targeting FER kinase. The subject is of high interest, and my comments - mainly on the structural aspects of the study - on the paper are listed below.

1. Format issues: Some parts of the text could use proofreading and grammar correction. The paper has 5 instances of "data not shown", and I would urge the authors to rethink this aspect. Some references are incomplete. The tables are listed as "in the following table" without numbering - is this really the journal style?

2. In the abstract and elsewhere, it is stated that microtubule bundling by CRMP-2 is "a previously unrecognised role". Fukata et al. 2002 at least studied this 15 years ago ("Strong assembly and bundle formation of microtubules were observed in the presence of purified native CRMP-2..." and Fig 3 in that paper), and it does not seem to be a novel idea.

3. The abstract (and other parts of the text) also states that Y479 phosphorylation results in significant conformational change. a) The conformational change is minor, a small turn in the last helix b) the effect is rather on oligomeric state c) The results do not show that phosphorylation causes this, but the phosphomimicking mutation.

4. Beginning of results: "...CRMP2 at Y32, a known modulator of..." This would mean Y32, not CRMP2, is known to modulate microtubule assembly - is this true?

5. The cosedimentation assay (Fig 1c) - quite a lot of CRMP2 goes into the pellet also without microtubules. How can we rule out that the more extensive pelleting of CRMP2 with microtubules is not caused by a simple crowding effect? This is a common concern in such cosedimentation assays.

6. Truncation mutants were used to show that the C terminus is important for bundling, but the results are not shown, so we do not get an idea at all.

7. Several Tyr-Phe mutants were used to generate non-phosphorylatable mutants. Are these proteins correctly folded? H bond interactions by the Tyr OH group may be important, and Phe is actually chemically rather different in this respect, especially if the side chain is buried.

8. When referring to ranges of amino acids, the expression "13-516 aa" is awkward. Please use e.g. "residues 13-516".

9. The authors throughout the text say how their structure, all the way to residue 516, is much longer than the earlier structures. However, only one chain has been built up to residue 507, and the other 4 are much shorter. The longest one is kept in place by a tight crystal contact. How these issues are described is misleading the reader, also in the discussion, where it is claimed that the structures include 26 more residues than earlier.

10. The discussion on Y479 phosphorylation and its possible effects on conformational changes/oligomeric state is very similar to that in Myllykoski et al. 2017 ("In the structure, Tyr479 is buried and involved in H-bonds through its hydroxyl group (Fig. 7c, d). Hence, the Y479F mutant used in functional studies (Varrin-Doyer et al. 2009) would likely be compromised in folding at least locally at this position. It can safely be said that Tyr479 phosphorylation requires major conformational change to occur; such a change could also affect e.g. the oligomeric state of CRMP-2 and the flexibility of the C-terminal region."), and for fairness, it would be good to cite this paper within this discussion.

11. The discussion mentions the relevance of the structural data to two modes of microtubule interaction: anchoring and bundling. It is not clear if and how this was actually shown in the paper.

12. Fig 3b legend says the mutation disrupts the helix and causes a steric clash. a) the helix is not disrupted b) the steric clash would only happen if the structure were a tetramer. This is confusing. The structural figures are also not very informative in general.

13. One major piece of novelty is the dimeric form of a CRMP, which to my knowledge has not earlier been seen. I am wondering why no figure shows this dimeric form, and why are its properties not discussed? Was it also validated in solution by e.g. SAXS? Such a form could also be important for other CRMP2 functional properties, such as described in recent papers on CRMP2 SUMOylation.

14. Supplementary Fig 1b. The CRMP2 preparation is really dirty, with several impurities. What experiments was this protein actually used for?

15. Methods - structure determination. Please give reference for PDB 2GSE. For solving the mutant structure, the authors searched with a monomer and say "single copy of the dimer was found" - this is incorrect as the search model was a monomer.

16. Crystallography table. Units for most parameters are missing. The number of significant digits for many listed parameters is too large. Clashescore - is this from Molprobit? Please list also the Molprobit percentile of each structure.

Response to reviewers

Reviewer 1:

This is a well executed study addressing the issue of controlling the state of microtubule dynamics in order to enhance the tumor responses to paclitaxel. Specifically the authors convincingly demonstrate that phosphorylation of the microtubule associated protein, CRMP2, by FER kinase, reduces its microtubule polymerizing activity, and its ability to induce microtubule elongation and to bundle microtubules. The authors identified by phosphopeptide mapping six phosphorylated sites among which two displayed structural and functional roles in this context. By solving the crystal structure of the wild type and the phosphomimetic mutants they propose a mechanism through which the phosphorylation impairs the tetramer conformation of CRMP2 thus impairing binding to microtubules. Notably one of this phosphorylation sites has been identified in a previous publication. The specific contribution of FER to this biological process is further demonstrated in this work by measuring the outcome of its inhibition and /or its reduced expression by siRNA, on the cytotoxic responses to paclitaxel. This was done in several cell lines and in xenograft mouse models of ovarian cancer. Altogether this is an interesting study which combines biochemical assays, protein structural studies, in vitro microtubule bundling assays, with cell based responses to paclitaxel, and might have a potential clinical implication in cancer.

We are thankful to the valuable comments of the reviewer and have now addressed them to the best of our efforts.

Comments and suggestions:

1. As mentioned by the authors, several Ser/Thr key kinases were previously shown to phosphorylate the C-terminal part of CRMP2 at other sites, which also block its ability to bind to tubulin. For example, Thr 514 phosphorylation by GSK-3 β . The authors should refer to these other functional phospho sites in the Discussion, and explain the difference/ common features concerning the outcome of these multiple kinases.

- i. We have added the following in the discussion in the revised version of the manuscript:

“CRMP2 is critical for axon formation by promoting microtubule assembly¹³. However, GSK-3 β and Rho kinase phosphorylate CRMP2 at T514 and S555, and inactivate it by impairing its association with tubulin dimers^{18,19} and inhibiting microtubule assembly. Our study reveals that FER phosphorylated CRMP2 at Y479 and Y499, reducing its microtubule bundling activity (Fig. 2). Though all these three kinases have negative impact on CRMP2’s microtubule assembly activity, FER kinase regulates microtubule assembly

via a different mechanism, compared with GSK-3 β and Rho kinase. Our study shows that phosphorylation at Y479 induces CRMP2 conformation change, impairing its tetramerization (Fig. 3), which is critical for its microtubule bundling function. However, how this conformation defect impairs CRMP2's activity should be investigated by resolving the structure of CRMP2-microtubule complex, using techniques such as cryo-EM'.

2. It is interesting that FER shares with FES kinase the phosphorylation on Y32. The importance of this site was excluded in this work since its substitution to phenyl alanine in the context of CRMP2-4F construct retained bundling activity and remained sensitive to FER. Yet the authors should test the possibility that a phospho mimicking mutation at this site might have an effect on its own or might synergize with the other two sites under some conditions.

- i. We have generated the Y32 phospho mimicking mutant proteins, including CRMP2-Y32E and CRMP2-Y32E-Y479E-Y499E. We performed the microtubule bundling assay *in vitro* using these proteins. The data showed that CRMP2-Y32E proteins had no reduced activity on bundling microtubules, compared with CRMP2 wild-type proteins (**Supplementary Fig. 4a**). This is further confirmed by the result that CRMP2-Y32E-Y479E-Y499E proteins and CRMP2-Y479E-Y499E proteins shared similar activity on bundling microtubules (**Supplementary Fig. 4b**). We have included these data in the revised version of the manuscript.

3. Is the Yes kinase expressed in the ovarian cell lines tested in this work? If positive what is the outcome of Yes knock down both at the cellular level and on Y479 phosphorylation? This should be examined. What is the range of specificity of the FER inhibitors used in his work?

- i. We are grateful to the reviewer for this important comment. We investigated the YES kinase expression level in multiple ovarian cancer cell lines. The result showed that it is expressed in most of the ovarian cancer cell lines (**Supplementary Fig. 3g**). However, YES kinase was hardly detected in the ovarian cancer cell line OVCA432, which is one of the cell models in our study (**Supplementary Fig. 3g**). We have included this data in the revised version of the manuscript.
- ii. To test the effect of loss of YES on pY479-CRMP2, we chose OVCAR3 cell line that had high expression of YES kinase and also expressed FER. The data revealed that the level of pY479-CRMP2 decreased in OVCAR3 cells with the depletion of either YES or FER kinases, compared with the cells with the non-targeting siRNA treatment (**Supplementary Fig. 3h,i**). However, the depletion of both kinases together did not further reduce pY479-CRMP2. This suggests that both kinases might be in the same pathway.
- iii. We (Nathanael S. Gray's lab) have previously reported that all the chemical compounds of TAE684, WZ-4-49-8, and TL-2-59 are able to target FER/FES kinase (Hellwig et al., 2012; Tan et al., 2015). These papers have been cited in the manuscript.

4. While the suppression of FER activity/expression affects the cytotoxic responses to paclitaxel in several ovarian cell lines, a statistically significant correlation between tumor responses to paclitaxel in patients and FER expression levels and/or Y479 phosphorylation levels is still missing. Thus, the reason for tumor resistance to paclitaxel is not clear yet and cannot be attributed exclusively to FER levels at this stage. This should be discussed.

- i. We have tested the correlation between the level of FER in multiple cell lines and response to paclitaxel and found that: FER expression level correlates with the IC50 value in ovarian cancer cell lines (**Supplementary Fig. 8a**).
- ii. Previous work has shown that there was a significant correlation between the level of FER and clinical outcome (progression free survival) in a large number of ovarian cancer patients (over 1000) who are routinely treated using paclitaxel. This work has now been referenced and highlighted in discussion
- iii. One limitation of this study is that the anti-pY479 antibody was not suitable for immunohistochemistry. Therefore, it was not possible to test the correlation between the level of FER and pY479 in patient samples. Therefore, as suggested by the reviewer, the following was added in the discussion:
“A limitation of this study is that the anti-pY479 antibody was not suitable for immunohistochemistry. Therefore, it was not possible to test the correlation between the level of FER and pY479-CRMP2 in patient samples. It is, therefore, not possible to test whether or not the intrinsic FER activity can correlate with paclitaxel resistance in ovarian cancer patients.”
- iv. Please note that in this study we have not claimed that FER can be used as a biomarker of paclitaxel response in cell lines or in patients. This was not the aim of the study. Instead we believe that in cancer cells that express FER, it is possible to enhance paclitaxel response by interfering with the kinase. We provide strong evidence that this occurs via enhanced microtubule bundling and stabilization. Please note the following in our introduction:
“We and others have previously shown that the state of microtubule stability in cancer cells prior to treatment, is an important determinant of the magnitude of paclitaxel-induced microtubule stabilization⁹⁻¹². However, whether microtubule dynamics manipulation can be exploited for enhancing paclitaxel cytotoxicity has remained untested.”

Reviewer 2:

We are grateful to the insightful comments from the reviewer and have now addressed them to the best of our ability.

Major:

1. The authors should include relevant literature supporting FER kinase as an oncogene. The currently cited reference (12) did not specifically describe FER as an oncogene.

- i. We have revised our statement to clarify that FER promotes metastasis in ovarian cancer, :

“FER kinase has been reported to promote ovarian cancer metastasis²⁰”.

2. If there is a lack of literature supporting FER expression in human ovarian malignancies, a detailed, large-scale expression study in ovarian tumour tissues and normal tissues should be performed. Association of expression with pathological or clinical parameters as well as the relationship between FER and/or CRMP2 expression and patient clinical outcome should be conducted.

- i. FER has been reported to promote ovarian cancer metastasis in a previous report (Fan et al., 2016), where the investigators performed a detailed and large-scale study of FER expression in human ovarian tumour tissues and normal tissues (more than 1000 samples in total). They revealed that FER expression was inversely correlated with progression-free survival. We have cited this literature in the revised manuscript.

3. Along the same lines, since an antibody against FER-phosphorylated-CRMP2 at the Y497 amino acid residue has been generated and the antibody specificity was assessed (e.g. Fig. 2e), it would be informative to assess the correlation between FER expression level and pY497 levels of CRMP2 on a panel of human tumor tissues and cell line models. Additionally, it would be important to use both in vitro and in vivo models to demonstrate whether CRMP2 pY497 levels can be inhibited by FER kinase inhibitor and, if yes, to what extent as compared to the FER knockdown approach. The goals of the above experiments are to provide additional supports for the presence and prevalence FER/CRMP2 regulatory axis in in vivo context.

- i. We analysed the FER expression level in multiple ovarian cancer cell lines by direct blotting using anti-FER antibody. To investigate pY479-CRMP2 level, we immunoprecipitated CRMP2 in multiple ovarian cancer cell lines, and then performed western blot using anti-phospho-Y479-CRMP2 antibody. We investigated the correlation between FER expression and pY479-CRMP2 in multiple ovarian cancer cell lines, and have included this data in the revised version of the manuscript. The result showed that FER expression level

correlates with pY479-CRMP2 level in multiple ovarian cancer cell lines ($r^2 = 0.924$) (**Supplementary Fig. 3e,f**).

- ii. We have now performed the *in vitro* kinase assay and used the cell model to test whether pY479-CRMP2 levels can be inhibited by FER kinase, TAE684. The results showed that TAE684 not only inhibit FER kinase activity to reduce the phosphorylation of CRMP2 in vitro using kinase assay (**Supplementary Fig. 3a**), but also decrease the phosphorylation of CRMP2 in ovarian cancer cells (**Supplementary Fig. 3d**).
- iii. One limitation of this study is that the anti-pY479 antibody was not suitable for immunohistochemistry or immunofluorescence. Therefore, it was not possible to test the correlation between the level of FER and pY479 in patient samples. Therefore, the following was added in the discussion:
“A limitation of this study is that the anti-pY479 antibody was not suitable for immunohistochemistry. Therefore, it was not possible to test the correlation between the level of FER and pY479-CRMP2 in patient samples. It is, therefore, not possible to test whether or not the intrinsic FER activity can correlate with paclitaxel resistance in ovarian cancer patients.”

4. To determine the therapeutic potential of the EFR targeting strategy, downregulation of EFR kinase was performed using siRNA or EFR inhibitors (e.g., TAE684, WZ-4-49-8) (Page 10, Fig. 5cd and Fig. 6); however, whether the approaches affect FER kinase activity was not presented or described. Since this is an important assessment to evaluate the specificity of the targeting strategy, such experiments should be performed.

- i. We (co-author Nathanael S. Gray) have previously reported that TAE684 and WZ-4-49-8 target FES and FER kinases (Hellwig et al., 2012). In this study, we have shown that TAE684 not only inhibit FER kinase activity to reduce the phosphorylation of CRMP2 in vitro using kinase assay (**Supplementary Fig. 3a**), but also decrease the phosphorylation of CRMP2 in ovarian cancer cells (**Supplementary Fig. 3d**).
- ii. We also provide the data that depletion of FER kinase using siRNA significantly reduces the phosphorylation of CRMP2 in ovarian cancer cells (**Fig. 2e**).

5. Because paclitaxel targets microtubules, their results suggested that the FER/CRMP2 axis may also regulate microtubule function, which is essential for mitosis. Can the authors detect alterations of the cell cycle and apoptosis when they modulate the FER/CRMP2 axis?

- i. We have investigated the function of FER kinase in the cell cycle in ovarian cancer cell line. The result showed that there were no alterations of the cell cycle in the presence or absence of FER in ovarian cancer cells, indicating that FER kinase was not critical for ovarian cancer cell cycle. We have shown the data (**Supplementary Fig. 7k**) in the revised version of the manuscript.

6. Figure 4b: Did pLX302:CRMP2-Y479E/Y499E show any significant change pertaining to resistance of microtubules to cold treatment compared with wild type CRMP2 and CRMP2-Y479F/Y499F? The authors did not mention this in the Results section.

- i. Yes, pLX302:CRMP2-Y479E/Y499E showed a significant reduction in resistance of microtubules to cold treatment compared with wild type CRMP2 ($p = 0.02$) and CRMP2-Y479F/Y499F ($p < 0.0001$). We have included these data in **Fig. 4b** in the revised version.

7. Fig. 5 and Supplementary Fig. 5: Does the endogenous expression of FER, CRMP2, or phospho-CRMP2 correlate with sensitivity to paclitaxel in this panel of ovarian cancer cells?

- i. We analysed the FER expression level in multiple ovarian cancer cell lines by direct blotting using anti-FER antibody. To investigate the paclitaxel sensitivity in multiple ovarian cancer cell lines, we performed cell proliferation assay for multiple ovarian cancer cell lines, treated with increasing concentration of paclitaxel. We analysed the IC50 value of individual ovarian cancer cell lines to determine the paclitaxel sensitivity in these cells. The data is shown in the following. The result showed that FER expression level correlates with the IC50 value in ovarian cancer cell lines ($r^2 = 0.804$) (**Supplementary Fig. 8a**). We have shown the data in the revised version of the manuscript.
- ii. Although the results showing in point (i) are persuasive, please note that in this study we have not claimed that FER can be used as a biomarker of paclitaxel response in cell lines or in patients. This was not the aim of the study. Instead we believe that in cancer cells that express FER, it is possible to enhance paclitaxel response by interfering with the kinase. We provide strong evidence that this occurs via enhanced microtubule bundling and stabilization. Please note the following in our introduction:
“We and others have previously shown that the state of microtubule stability in cancer cells prior to treatment, is an important determinant of the magnitude of paclitaxel-induced microtubule stabilization⁹⁻¹². However, whether microtubule dynamics manipulation can be exploited for enhancing paclitaxel cytotoxicity has remained untested”

8. Methods section on page 30 – Was the tumor xenograft study performed on C57BL/6 mice or on immunodeficient mice?

- i. Yes, the tumor xenograft study was performed on nude immunodeficient C57BL/6 mice. This is now clarified in the Methods section.

9. The second paragraph on page 9 (see below) is confusing. Was FER expressed in ovarian cancer cell lines or was un-detectable in ovarian cancer cell lines? “This was supported by the observation that FER was ubiquitously expressed in ovarian cancer cell lines (Fig. 5a) and strongly expressed in more than a third of 130 high grade serous ovarian cancers (HGSOCs) (Supplementary Fig. 5a,b). In contrast, FES was undetectable at protein (data not shown) or mRNA level in ovarian cancer cell lines (Supplementary Fig. 5c).”

- i. We have changed the writing “In contrast, FES was undetectable at protein (data not shown) or mRNA level in ovarian cancer cell lines (**Supplementary Fig. 5c**)” to “*In contrast, FES, the only other family member of FER, was undetectable at protein (data not shown) or mRNA level in ovarian cancer cell lines (**Supplementary Fig. 7c**)*”. This has been included in the revised version of the manuscript.

10. There are some issues pertaining to the animal study (Fig. 6 and supplementary Fig 6). To help evaluating the specificity of the observed *in vivo* phenotypes, the following experiments should be performed:

(1) The knockdown efficiency of these nanoliposomes should be described. In addition, the distribution of DOPC-nanoliposomes in mouse body and whether they selectively target tumour tissues should be demonstrated. Furthermore, their effects on the mouse body weight should be measured.

- i. We have shown the data in **Supplementary Fig.8o**. The results showed a significant reduction in FER protein (> 50% reduction, $p < 0.001$).
- ii. We have previously published the details of the distribution of the DOPC-nanoliposomes in mouse body. We have now referenced the studies below in the methods section:

Landen CN, Chavez-Reyes A, Bucana C, Schmandt R, Deavers MT, Lopez-Berestein G, Sood AK (2005). Therapeutic EphA2 gene targeting by *in vivo* liposomal siRNA delivery. *Cancer Research*.

Pecot CV, Rupaimoole R, Yang D, Akbani R, Ivan C, Lu C, Wu S, Han HD, Shah MY, Rodriguez-Aguayo C, Bottsford-Miller J, Liu Y, Kim SB, Unruh A, Gonzalez-Villasana V, Huang L, Zand B, Moreno-Smith M, Mangala LS, Taylor M, Dalton HJ, Sehgal V, Wen Y, Kang Y, Baggerly KA, Lee J-S, Ram PT, Ravoori MK, Kundra V, Zhang X, Ali-Fehmi R, Gonzalez-Angulo AM, Massion PP, Calin GA, Lopez-Berestein G, Zhang W, Sood AK (2013). Tumor angiogenesis regulation by the miR-200 family. *Nature Communications*.

- iii. We have included the mouse body weight in the revised version of manuscript (**Supplementary Fig.8n**).

(2) Detailed information about the preparation, dosage, and treatment cycles of

DOPC-nanliposomes should be provided in Materials and Methods.

- i. We have included these information in Materials and Methods in the new revision of the manuscript. It states that:

“siRNA for in vivo delivery was incorporated into 1,2-dioleoyl-sn-glycero-3-phosphocholine (DOPC; Avanti Polar Lipids, Inc.) as described previously³⁵. Briefly, 5 µg of siRNA and DOPC were mixed in the presence of excess tertiary butanol at a ratio of 1:10 (w/w). After addition of tween-20 (1:19 tween 20:siRNA/DOPC), the mixture was frozen in an acetone-dry ice bath and lyophilized. Liposomes were reconstituted with 200 µL (to achieve a 200 µg/kg concentration) phosphate-buffered saline (PBS) and administered into mice intraperitoneally. Mice were treated for 6 weeks, biweekly to have siRNA bioavailability and sustained silencing effect”.

(3) Since an ovarian cancer cell line with engineered expression of luciferase reporter was employed in the animal study, tumor growth kinetics measured by in vivo bioluminescent images should be demonstrated in this report.

Unfortunately, this was not performed in this study. This was an oversight. We feel that we provided sufficient alternative evidence by showing tumour weight, counts of tumour nodules and also now showing the mouse body weight.

(4) If FER inhibitor is proven to be safe for clinical use, it can be included as another agent for a proof-of-principal study.

To our knowledge, the current known FER inhibitors have not been tested for clinical use. We are working on identifying and developing suitable FER inhibitors for future translation into clinical application.

11. Quantification and statistical analysis - many of the studies are comparing more than two group means; t-test is inappropriate to be used. The authors should consider using one-way ANOVA when comparing means of more than two groups.

- i. We have performed the one-way ANOVA analysis for the studies, which are comparing more than two groups. These data have been included in the revised version of the manuscript.

Minor:

• Please specify how microtubule elongation and microtubule width were measured and calculated.

- i. We have included the description of how to measure and calculate microtubule elongation and microtubule width in the revised manuscript. It states that:

*“For microtubule elongation rate calculation, we measured the length of a single microtubule at starting point (T_1 , second), which gave the length (L_1 , μm). Then we selected the ending point (T_2 , second), and measured the length of the same single microtubule (L_2 , μm). The length of microtubules was measured using Image J software. More than 25 microtubules were randomly selected to be measured. Microtubule elongation rate was calculated using the following equation: $(L_2 - L_1)/[(T_2 - T_1)*60]$.”*

For microtubule bundle width calculation, we measured the maximum width of each microtubule, using Image J software. More than 150 microtubules (more than 8 μm of length) were randomly selected to be measured”.

• Based on the description, it is not clear how many different FER inhibitors were used for this study. The authors sometimes described TAE684 and WZ-4-49-8, and at other times described TAE684 and TL-2-59.

- i. We have now included the data of colony formation assay using WZ-4-49-8 and paclitaxel (**Supplementary Fig. 8j,k**) in the revised version of the manuscript in addition to the data that were previously presented in the original submission for inhibitors TAE684 and TL-2-59.
- ii. All the chemical compounds of TAE684, WZ-4-49-8, and TL-2-59 are inhibitors of FER/FES kinase, which have been reported in the previous literatures (Hellwig et al., 2012; Tan et al., 2015). In our study, TAE684 and WZ-4-49-8 were used to inhibit FER kinase in the assays to investigate microtubule stability or microtubule length elongation in ovarian cancer cells.

• Is the image shown in Fig 5C generated from FER 01 siRNA or FER 02 siRNA? Please include images for both siRNAs.

- i. The image shown in Fig. 5C was generated from FER 01 siRNA. We have included the images for both siRNAs (**Fig. 5c**) in the revised version of the manuscript.

• In terms of measuring microtubule dynamics in live cells, the authors should consider tracking fluorescently labelled plus-end-tracking proteins over time, performing fluorescence recovery after photobleaching (FRAP) on cells transfected with plasmid encoding fluorescently labelled proteins, or method mentioned in the following paper: PMID 21773917.

- i. We have not been investigating microtubule dynamics in live cells. All the cells stained with anti-glu tubulin antibody or anti- α tubulin antibody were fixed using 4% paraformaldehyde before staining.

To avoid being confusing, we re-write one of the sub-titles in the part of “Results” (on Page 10). We change to the sub-title “FER Regulates Microtubule Dynamics via CRMP2 in Ovarian Cancer Cells” to “*FER Regulates Microtubule Stability via CRMP2 in Ovarian Cancer Cells*” in the revised version of the manuscript.

Reviewer 3:

Zheng et al. "Tuning microtubule dynamics..."

The paper discusses the interplay of CRMP2 phosphorylation by FER kinase, CRMP2 structure, and microtubule dynamics, in order to pave new ways for cancer therapeutics targeting FER kinase. The subject is of high interest, and my comments - mainly on the structural aspects of the study - on the paper are listed below.

We appreciate the valuable comments of the reviewer and have now addressed them to the best of our efforts.

1. Format issues: Some parts of the text could use proofreading and grammar correction. The paper has 5 instances of "data not shown", and I would urge the authors to rethink this aspect. Some references are incomplete. The tables are listed as "in the following table" without numbering - is this really the journal style?

- i. We thank the reviewer for noticing the grammatical errors and have now proof-read the article to the best of our abilities.
- ii. We are grateful to the reviewer to notice that the paper has 5 instances of "data not shown". We have shown all the data for these 5 instances except the FES protein expression level (on page 9) as it was unable to be detected. We have included the data for other 4 instances in the revised version of the manuscript. They are shown as **Supplementary Fig. 2a, Supplementary Fig. 2b,c, Supplementary Fig. 7f, and Supplementary Fig. 7h.**
- iii. We believe that this has been completed in the revised version of the new manuscript. However, we do appreciate that the reviewer will respond again in case of some additional references needed to be cited.
- iv. In terms of the tables were listed without numbering, we have now corrected this to state the table numbers in the revised version of the manuscript.

2. In the abstract and elsewhere, it is stated that microtubule bundling by CRMP-2 is "a previously unrecognised role". Fukata et al. 2002 at least studied this 15 years ago ("Strong assembly and bundle formation of microtubules were observed in the presence of purified native CRMP-2..." and Fig 3 in that paper), and it does not seem to be a novel idea.

- i. The investigators in 2002 revealed that CRMP2 binds to tubulin dimer via central domain (residues 323-381) to promote microtubule assembly, and indeed the investigators mentioned the microtubule bundle formation in the presence of CRMP2 proteins in their study. Therefore, we agree that it would not be a new concept that CRMP2 is able to bundle microtubules in our study. However, our investigation is still of novelty in terms of CRMP2's function in microtubule bundling due to the following reasons:
 - a. The previous study (Fukata et al. 2002) showed that the central domain of CRMP2 is pivotal for promoting microtubule assembly. However, we

find that the carboxy-terminus is critical in CRMP2's microtubule bundling activity.

- b. In the previous report (Fukata et al. 2002), microtubule bundling formation was observed via the microtubule polymerization assay using tubulin dimers. We performed microtubule bundling assay with single pre-formed microtubules but not tubulin dimers. Our data revealed that CRMP2 interacts with single microtubules to initiate microtubule bundling, which is independent on tubulin dimer mediated microtubule assembly. Therefore, our data clarify that CRMP2 bundles microtubule on single microtubules.

We re-organized the writing in the revised version of the manuscript.

3. The abstract (and other parts of the text) also states that Y479 phosphorylation results in significant conformational change. a) The conformational change is minor, a small turn in the last helix b) the effect is rather on oligomeric state c) The results do not show that phosphorylation causes this, but the phosphomimicking mutation.

- i. We agree with the reviewer's comments and have now changed the "conformational change" to specify that the effect is rather on the oligomerization state of CRMP2. Also, we now specifically state that this effect is because of the mutation from Y479 to E479 which mimics phosphorylation.

4. Beginning of results: "...CRMP2 at Y32, a known modulator of..." This would mean Y32, not CRMP2, is known to modulate microtubule assembly - is this true?

- i. We have change the writing as "*In addition, previous reports found that FES, the only other family member of FER, is able to phosphorylate Y32 of CRMP2. CRMP2 has been reported to be a known modulator of microtubule assembly*". This has been included in the revised version of the manuscript.

5. The cosedimentation assay (Fig 1c) - quite a lot of CRMP2 goes into the pellet also without microtubules. How can we rule out that the more extensive pelleting of CRMP2 with microtubules is not caused by a simple crowding effect? This is a common concern in such cosedimentation assays.

- i. We agree that some part of CRMP2 proteins would appear in the pellet in the absence of microtubules when performing low speed cosedimentation (13 000 g at room temperature) (**Fig. 1c**). This might be due to the reason that CRMP2 proteins were exposed in the microtubules buffer condition, and were not that much stable as they were in the CRMP2 protein purification buffer.
- ii. However, we still believe that it is safe and reasonable to make the conclusion that CRMP2 is able to associate with microtubules in this study, by presenting the following evidences:
 - a. We have now measure the ratio between the soluble fraction and the pellet fraction in the absence or presence of microtubules. There is a

280 ($19.6 / 0.07 = 280$) fold difference in the ratio between in the absence of microtubules and presence of microtubules. This has now been added in **Fig. 1c**.

- b. Immunofluorescence staining using anti-CRMP2 antibody clearly showed that CRMP2 proteins bound to microtubules *in vitro* (**Fig. 1d**).
- c. In cells, we demonstrate that CRMP2 binds to microtubules of the mitotic spindle by overexpressing CRMP2 proteins. Moreover, the phosphomimic CRMP2 proteins (mimicking the phosphorylation at Y479 of CRMP2 by FER kinase) have the significantly reduced binding activity to mitotic microtubules, further supporting that wild-type CRMP2 associates with microtubules (**Fig. 4a and Supplementary Fig. 6a**).

6. Truncation mutants were used to show that the C terminus is important for bundling, but the results are not shown, so we do not get an idea at all.

- i. We have included these data in the revised version of the new manuscript. The results indicated that CRMP2 (residue 13-490) proteins have no activity on microtubule bundling activity. However, the CRMP2 truncation mutant proteins showed increased microtubule bundling activity with the increasing length of the carboxy-terminus of CRMP2 (**Supplementary Fig. 2b,c**).

7. Several Tyr-Phe mutants were used to generate non-phosphorylatable mutants. Are these proteins correctly folded? H bond interactions by the Tyr OH group may be important, and Phe is actually chemically rather different in this respect, especially if the side chain is buried.

- i. We have now performed Circular Dichroism Spectrophotometry on the full length CRMP2 wild-type and on the full length CRMP2-6F to show that mutating the 6 FER phosphorylation sites on CRMP2 from Y to F has no effect on the secondary structures in the mutants (**Supplementary Fig. 3j**), from which we conclude that the mutants are correctly folded. This is also written in the results section, under the sub-heading "*FER phosphorylates CRMP2 at Y479 and Y499*", in the last paragraph as:

"We also confirmed that the mutation from Y to F does not affect the secondary structures of the various non-phosphorylatable mutants, by performing Circular Dichroism (CD) spectrophotometry on CRMP2-6F (all six FER phosphorylation sites mutated) and CRMP2 wild-type (Supplementary Fig. 3j)."

8. When referring to ranges of amino acids, the expression "13-516 aa" is awkward. Please use e.g. "residues 13-516".

- i. "13-516 aa" has now been replaced with "residues 13-516" throughout the manuscript.

9. The authors throughout the text say how their structure, all the way to residue 516, is much longer than the earlier structures. However, only one chain has been built up to residue 507, and the other 4 are much shorter. The longest one is kept in place by a tight crystal contact. How these issues are described is misleading the reader, also in the discussion, where it is claimed that the structures include 26 more residues than earlier.

- i. We agree with the comments of the reviewer and have now tried to explain this issue better. In the Results section, under the sub-heading “*Structural Evidence for Disruption of Inter-Molecular Interactions upon Phosphorylation of CRMP2*”, we now write :

“Our wild-type CRMP2 protein construct has 26 extra residues (residues 491-516) at the carboxy-terminus as compared to the constructs of the previously solved structures (PDB ID 2GSE and 5LXX). However, in the crystal structure, the electron density is only best visible up to residue 507, in one out of the 4 chains, suggesting flexibility in this region. The residues starting from 490 onwards seem to be interacting with their dimer-partner either via hydrogen bonds or salt bridges (Fig. 3a, left panel). This interaction was not seen in previous structures because the constructs were too short. We speculate that the flexible carboxy-terminus is required for the CRMP2 to probe and associate with the microtubules. The fact that Y499, is located in this region (Fig. 3a, left panel) and that our previous experiments with the carboxy-terminal truncation mutants showed this region to be important for bundling microtubules, we hypothesized that if Y499 is phosphorylated, the carboxy-terminus will not be able to bind the microtubules efficiently”.

10. The discussion on Y479 phosphorylation and its possible effects on conformational changes/oligomeric state is very similar to that in Myllykoski et al. 2017 (“In the structure, Tyr479 is buried and involved in H-bonds through its hydroxyl group (Fig. 7c, d). Hence, the Y479F mutant used in functional studies (Varrin-Doyer et al. 2009) would likely be compromised in folding at least locally at this position. It can safely be said that Tyr479 phosphorylation requires major conformational change to occur; such a change could also affect e.g. the oligomeric state of CRMP-2 and the flexibility of the C-terminal region.”), and for fairness, it would be good to cite this paper within this discussion.

- i. We agree with the feedback and have now duly cited that paper within the discussion.

11. The discussion mentions the relevance of the structural data to two modes of microtubule interaction: anchoring and bundling. It is not clear if and how this was actually shown in the paper.

- i. We have re-written the discussion about the relevance of the structural data to interaction with microtubules. Followed by further discussion in the Discussion section of the revised version of the new manuscript, it states that:

“The new structural information has enabled us to speculate that both the intact tetramer conformation and the carboxy-terminus of CRMP2 are pivotal for CRMP2’s function to bundle microtubules”.

12. Fig 3b legend says the mutation disrupts the helix and causes a steric clash. a) the helix is not disrupted b) the steric clash would only happen if the structure were a tetramer. This is confusing. The structural figures are also not very informative in general.

- i. The structure figure (**Fig. 3**) has now been improved to show: Important features of the tetramer structure in Panel a, dimer structure in Panel b, and the superposition between the tetramer and the dimer structure to show the effect of mutation on the tetramerization of CRMP2 Panel c. The legend for the figure has also been changed to

“Phosphorylation of CRMP2 at Y479 Induces Oligomerization Changes in CRMP2 Structure

(a) (Left panel) Surface representation of a tetramer of CRMP2 (residues 13-516) showing the dimerization interface. Visible carboxy-terminal (C’) extensions of each chain are shown as ribbons. Reader visible Y499 are labelled and shown in cyan. (Right panel) Tetramerization interface between the dimers is shown. The position of Y479 at the interface is also highlighted in blue. Zoom in shows the details of this interface comprising the carboxy-terminal helix of chain A and residues 372-379 of the neighboring chain B.

(b) Crystal structure of the phosphomimetic mutant CRMP2-2E. (Left panel) Position of E479 is highlighted. (Right panel) Position of SUMOylation site, K374, is highlighted.

(c) Superposition of CRMP2 wild-type tetramer structure and CRMP2-2E phosphomimetic mutant dimer structure. Zoom in showing key differences at the tetramerization interface: Chain A (green) of CRMP2 (residues 13-516) has the Y479 (blue) at the carboxy-terminal helix which interacts with the loop (gray) of chain B (orange). CRMP2-2E mutant (red) has E479 in the position. The mutation introduces positive charge in the hydrophobic cavity (See also Supplementary Fig. 3) causing the carboxy-terminal helix to unwind and the tetramer to break.”

13. One major piece of novelty is the dimeric form of a CRMP, which to my knowledge has not earlier been seen. I am wondering why no figure shows this dimeric form, and why are its properties not discussed? Was it also validated in solution by e.g. SAXS? Such a form could also be important for other CRMP2 functional properties, such as described in recent papers on CRMP2 SUMOylation.

- i. **Fig. 3, Panel B**, now shows the dimer structure. We have also validated this dimer in solution with analytical gel filtration and show the data in **Supplementary Fig. 5b**. Accordingly, in the results section under the sub-

heading “*Structural Evidence for Disruption of Inter-Molecular Interactions upon Phosphorylation of CRMP2*”, at the end of paragraph 2, we write

“*Analytical gel filtration further validated that phosphomimetic mutant is a dimer in solution as compared to the tetrameric wild-type CRMP2 (Supplementary Fig. 5b)*”.

We further discuss the implications of this dimeric form on the regulation of biological functions of CRMP2 in the Discussion part by writing “*Our work reports the existence of CRMP2 as a dimer upon phosphorylation, which to our knowledge, has not been observed before. This is especially interesting, considering the recent study reporting an interplay between phosphorylation and SUMOylation regulating the biological functions of CRMP2³⁰. The investigators show the existence of structurally conserved SUMOylation site, K374, which remains buried at the tetrameric interface of CRMP2. They further predicted that this site would be solvent-accessible, if interface was disrupted. Our phosphomimetic mutant structure, indeed, is in accordance with this prediction and clearly shows that K374 is available for SUMOylation, in the dimeric form (Fig. 3b right panel)*.”

14. Supplementary Fig 1b. The CRMP2 preparation is really dirty, with several impurities. What experiments was this protein actually used for?

- i. The purified CRMP2 proteins look dirty with several impurities due to the following reasons: a, we loaded too much of these proteins for SDS-PAGE running; b, we run this protein sample before gel filtration. We replace the old CRMP2 preparation image with the new one, which was obtained using the purified CRMP2 proteins after gel filtration. We have included this in the revised version of the manuscript (**Supplementary Fig. 1b**). These purified proteins were used for kinase assay *in vitro* and microtubule bundling assays.

15. Methods - structure determination. Please give reference for PDB 2GSE. For solving the mutant structure, the authors searched with a monomer and say “single copy of the dimer was found” - this is incorrect as the search model was a monomer.

- i. We have now properly cited the reference for the PDB 2GSE and have also corrected the statement in the method section to

“*The structure was phased using CCP4- Phaser³⁸ by providing a monomer from CRMP2 tetramer (residues 13-490, PDB: 2GSE)²⁵ as the template. Two copies of the monomer were found in the asymmetric unit. The structure was refined using CCP4-Refmac5³⁹. The data collection and refinement statistics are provided in Table 4.*”

16. Crystallography table. Units for most parameters are missing. The number of significant digits for many listed parameters is too large. Clashscore - is this from Molprobit? Please list also the Molprobit percentile of each structure.

- i. The units are now mentioned for all the applicable parameters. The number of significant digits is also reduced. The clashscore is from Molprobit and we also now list the Molprobit percentile for each structure.

REVIEWERS' COMMENTS:

Reviewer #1 (Remarks to the Author):

Evaluation of the response to the four points raised by the reviewer:

1. The mechanistic differences between the outcome of CRMP2 phosphorylation by FER and the two other kinases that target other sites (GSK-3 β and Rho kinase) are clarified by the authors. Are the two kinases expressed in the ovarian cell lines tested in this work? Should they also be considered in future studies as targets for inhibition to increase paclitaxel responses? This part is not discussed.

2. OK

3. It is now shown that the knock down of YES also results in reduction of CRMP2 phosphorylation at Y479 in the OVCAR3 cell line. As the double KO FER/YES fails to show additive effects on pY479 phosphorylation, the authors suggest that they operate in the same pathway. This point is not clear. Did the authors measure the effect of YES knock down on paclitaxel sensitivity in OVCAR3? Is YES another target to be studied in the context of paclitaxel responses in ovarian cancer cells?

A minor point- the blot in Supplementary Fig, 3h does not reflect the statistical data shown in 3i- it is better to show another blot from the three replicates used in Supp 3i.

4. The information concerning a possible correlation between the FER expression levels and paclitaxel responses in patients is still missing. The author quote a paper (ref 20) which documented inverse correlation between FER expression levels and progression free survival. Yet in this data set only around 15% of patients were treated with paclitaxel (<http://www.kmplot.com>). So clinical data which support the author's suggestion that manipulation of microtubule dynamic instability, specifically through FER inhibition, might increase cytotoxic responses of patients to paclitaxel is missing. The authors should be more careful in the clinical interpretation of their data.

Reviewer #2 (Remarks to the Author):

The authors have adequately addressed the questions and concerns. There is no further concerns.

Reviewer #3 (Remarks to the Author):

I am glad to see the authors have carefully responded to all my earlier comments and I have no further concerns, apart from the minor issue given below. Clearly, a significant amount of good work has been done to reach this revision of the manuscript.

In table 4, the unit for the three data processing R factors as well as CC1/2 is given as (%), but the numbers are not in % (i.e. they should be multiplied 100-fold or the unit should be removed). In the same table, the unit for average B factors is still missing (\AA^2).

Response to Reviewers

Reviewer 1 (Remarks to the Author):

Evaluation of the response to the four points raised by the reviewer:

1. The mechanistic differences between the outcome of CRMP2 phosphorylation by FER and the two other kinases that target other sites (GSK-3 β and Rho kinase) are clarified by the authors. Are the two kinases expressed in the ovarian cell lines tested in this work? Should they also be considered in future studies as targets for inhibition to increase paclitaxel responses? This part is not discussed.

- i. Previous reports have shown that GSK-3 β and Rho kinase are expressed in the ovarian cancer cell lines tested in this work. This has now been stated in the discussion:

“Though all these three kinases have negative impact on CRMP2’s microtubule assembly activity, FER kinase regulates microtubule assembly via a different mechanism, compared with GSK-3 β and Rho kinase that are both known to be expressed in ovarian cancer cells^{30,31”}.

- ii. While the two kinases are highly likely to modify paclitaxel response, their clinical utility is uncertain. Both kinases are involved in a wide range of regulatory pathways in the cell. Therefore, interfering with their activity is likely to have toxic effects.

2. It is now shown that the knock down of YES also results in reduction of CRMP2 phosphorylation at Y479 in the OVCAR3 cell line. As the double KO FER/YES fails to show additive effects on pY479 phosphorylation, the authors suggest that they operate in the same pathway. This point is not clear. Did the authors measure the effect of YES knock down on paclitaxel sensitivity in OVCAR3? Is YES another target to be studied in the context of paclitaxel responses in ovarian cancer cells?

A minor point- the blot in Supplementary Fig, 3h does not reflect the statistical data shown in 3i- it is better to show another blot from the three replicates used in Supp 3i.

- i. We do acknowledge that the mechanistic basis of the interaction between YES and FER have not been addressed in this study. We have stated that this will be an interesting question for future studies:

*“In the future, it will be interesting to investigate whether FER and YES co-regulate microtubule stability in ovarian cancers in terms of modulating CRMP2 activity on microtubules as YES is expressed in some ovarian cancer cell lines and also involved in the phosphorylation at Y479 of CRMP2 in the ovarian cancer cell line (**Supplementary Fig. 3g-i**).”*

- ii. We have also added another sentence about the possible role of inhibiting YES in paclitaxel sensitization:

“It will also be of interest to test whether targeting YES kinase can enhance paclitaxel response.”

- iii. Please note that the data in **Supplementary Figure 3i** include normalized values that take into account the total level of CRMP2. In immunoprecipitation experiments, the latter will inevitably vary between different conditions. Therefore, quantification of band intensity is very important as the appearance of the blot may not reflect true biological variations. To make this point clearer, we have now shown the quantification value (normalized pY479-CRMP2 level) in **Supplementary Fig. 3h** in the revised version of the manuscript.

4. The information concerning a possible correlation between the FER expression levels and paclitaxel responses in patients is still missing. The author quote a paper (ref 20) which documented inverse correlation between FER expression levels and progression free survival. Yet in this data set only around 15% of patients were treated with paclitaxel (<http://www.kmplot.com>). So clinical data which support the author’s suggestion that manipulation of microtubule dynamic instability, specifically through FER inhibition, might increase cytotoxic responses of patients to paclitaxel is missing. The authors should be more careful in the clinical interpretation of their data.

- i. Thank you for pointing out the website <http://www.kmplot.com> to our attention. The website for ovarian cancer patients: <http://kmplot.com/analysis/index.php?p=service&cancer=ovar> says that the number of ovarian cancer patients who received chemotherapy containing paclitaxel is 248 which equates roughly to about 15% of cases. However, please note that Taxol is the Bristol Myers Squibb “tradename” for paclitaxel. This article <http://www.nature.com/articles/375432a0> and comments on it provide an interesting read about the history of how the name was given. But of relevance to our work, the website that is pointed out by the reviewer also mentions that 821 patients (~55%) received “Taxol”-containing chemotherapy. So, in total about 70% of patients may have received paclitaxel-containing chemotherapy. In addition, please note that the website mentions that there 108 patients (~7%) received the paclitaxel analogue Docetaxel that is also a microtubule stabilizing agent that works in an identical way to paclitaxel on microtubules. Please note that this high percentage of patients receiving taxane-containing chemotherapy in ovarian cancer is not surprising as this has been the standard of care in managing ovarian cancer patients for many years.

Reviewer #2 (Remarks to the Author):

The authors have adequately addressed the questions and concerns. There is no further concerns.

- i. Thank you.

Reviewer #3 (Remarks to the Author):

I am glad to see the authors have carefully responded to all my earlier comments and I have no further concerns, apart from the minor issue given below. Clearly, a significant amount of good work has been done to reach this revision of the manuscript.

In table 4, the unit for the three data processing R factors as well as CC1/2 is given as (%), but the numbers are not in % (i.e. they should be multiplied 100-fold or the unit should be removed). In the same table, the unit for average B factors is still missing (\AA^2).

- i. We are thankful to the reviewer's comment. We have addressed the minor issue in the revised version of the manuscript. It is **Supplementary Table 4** in the revised version of the manuscript.